METHODS AND RESOURCES

# DISSeCT: An unsupervised framework for high-resolution mapping of rodent behavior using inertial sensors

Romain Fayat[1], Marie Sarraudy[1], Clément Léna[1], Daniela Popa[1], Pierre Latouche[2], Guillaume P. Dugué [1]*

**1** Institut de Biologie de l'École Normale Supérieure, École Normale Supérieure, CNRS, INSERM, Université PSL, Paris, France, **2** Laboratoire de Mathématiques Blaise Pascal, Institut Universitaire de France, Université Clermont Auvergne, CNRS, Aubière, France

* guillaume.dugue@cnrs.fr

## Abstract

Decomposing behavior into elementary components remains a central challenge in computational neuroethology. The current standard in laboratory animals involves multi-view video tracking, which, while providing unparalleled access to full-body kinematics, imposes environmental constraints, is data-intensive, and has limited scalability. We present an alternative approach using inertial sensors, which capture high-resolution, environment-independent, compact 3D kinematic data, and are commonly integrated into rodent neurophysiological devices. Our analysis pipeline leverages unsupervised, computationally efficient change-point detection to break down inertial time series into variable-length, statistically homogeneous segments. These segments are then grouped into candidate behavioral motifs through high-dimensional, model-based probabilistic clustering. We demonstrate that this approach achieves detailed rodent behavioral mapping using head inertial data. Identified motifs, corroborated by video recordings, include orienting movements, grooming components, locomotion, and olfactory exploration. Higher-order behavioral structures can be accessed by applying a categorical hidden Markov model to the motif sequence. Additionally, our pipeline detects both overt and subtle motor changes in a mouse model of Parkinson's disease and levodopa-induced dyskinesia, highlighting its utility for behavioral phenotyping. This methodology offers the possibility of conducting high-resolution, observer-unbiased behavioral analysis at minimal computational cost from easily scalable and environmentally unconstrained recordings.

## Introduction

In animal studies, investigations into brain-behavior relationships have typically relied on constrained experimental environments with specialized instruments designed to monitor predefined actions, such as lever presses or nose pokes [1]. Advancing toward more flexible and ethologically relevant experimental designs requires innovative methods for continuously tracking animal activities over time and objectively parsing the underlying

**Data availability statement:** The IMU and 3D pose datasets supporting this study are publicly

available at https://doi.org/10.25493/8ATR-V87. The code of DISSeCT is available at https://zenodo.org/records/17077913. The code of INSpECT, a companion software used to examine and label video snippets, is available at https://zenodo.org/records/17077911.

**Funding:** This work was supported by the Agence Nationale de la Recherche (ANR-20-CE37-0016 to GD; ANR-19-CE37-0007 to DP) and the Fondation pour la Recherche Medicale (FRM-EQU202103012770 to CL). The funders had no role in study design, data collection and analysis, decision to publish, or preparation of the manuscript. Neither of the funders has played any role in the study design, data collection and analysis, decision to publish, or preparation of the manuscript.

**Competing interests:** The authors have declared that no competing interests exist.

action sequences [2,3]. Progress in this field has been largely driven by computer vision techniques [4,5], particularly through the use of convolutional neural networks to detect user-defined anatomical landmarks, or keypoints, in images of behaving animals [6–8]. Applied to multi-view video recordings, keypoint detection can be used to compute 3D pose estimates [9–14], which can then serve as input data for machine learning-based behavioral mapping frameworks [13,15–18].

However, these approaches come with several limitations that impose significant constraints on how they can be effectively deployed. Keypoint detection is particularly vulnerable to occlusions, which occur when body parts are obscured by environmental elements, other individuals, or when animals adopt contorted postures. Additionally, the estimation of keypoint localization is inherently jittery, reducing the spatio-temporal resolution of this technique and complicating subsequent data analysis [12,17]. In larger environments, the trade-off between image resolution and field-of-view leads to reduced spatial accuracy unless a complex, self-orienting video tracking system is employed [19]. Enhancing image resolution and increasing frame rate to minimize motion blur can produce sharper measurements but results in larger data volumes. 3D pose estimation presents further challenges, including the need for meticulous camera calibration, a strategy to mitigate potential frame drops, and the management of larger data loads, which are often incompatible with long-term recordings.

Wearable inertial measurement units (IMUs), which directly report the instantaneous 3D angular velocity and acceleration of specific body parts, provide a compelling alternative for monitoring behavior. These sensors produce small data files compatible with longitudinal onboard data logging, accounting for their widespread adoption among field ethologists [20, 21], animal welfare scientists [22], biomechanists [23], and clinicians [24,25]. Due to their high precision, exquisite temporal resolution, and compact, lightweight form factors, IMUs are also increasingly favored by physiologists for tracking rodent head [26–45], limb [46,47], and body [48–52] movements. Being environment-agnostic, these devices are ideally suited for transitioning toward more naturalistic experimental settings involving large, complex, or crowded setups. Reflecting their growing popularity in the field, IMUs are now routinely integrated into both commercial and open source head-mounted electrophysiological and imaging devices [53–55]. Despite these advantages, inertial data have rarely been used as a main source of information for automating behavioral analysis in experimental animal models [26, 43,51], in contrast with IMU-based approaches in human activity recognition [56], animal health monitoring [57], and ethological research [58]. Yet, the high sensitivity, data efficiency, and versatility of IMUs are well-suited for developing advanced analysis frameworks aimed at unraveling the architecture of naturalistic behavior, offering the enhanced resolution required by computational neuroethology [2] and deep behavioral phenotyping [59].

To unlock the potential of inertial recordings for mapping rodent behavior, we developed a fully unsupervised analysis pipeline called DISSeCT (Decomposing Inertial Sequences by Segmentation and Clustering in Tandem), which parses IMU signals into discrete activity motifs. Our pipeline employs a two-step process: first, transitions between statistically homogeneous segments are detected using an optimal, computationally efficient change-point detection method, taking full advantage of the low noise and high sampling rate of IMU recordings. The resulting segments are then grouped into candidate behavioral motifs using high-dimensional Gaussian mixture modeling. We ran our pipeline on head inertial recordings obtained from freely moving rodents using both wired and wireless 6-axis digital IMUs, leveraging the fact that these animals rely heavily on head movements to explore their environment [31,37,60] and that many of their activities exhibit distinct head inertial signatures [26,30,43,61]. We show that DISSeCT effectively decomposes inertial data into meaningful activity motifs, validated by concurrent video recordings, parsing the full repertoire

of orienting movements, isolating rearing and locomotion episodes, distinguishing the main components of self-grooming, and highlighting episodes of olfactory exploration. Using a mouse model of Parkinson's disease and levodopa-induced dyskinesia, we also illustrate how our pipeline can be used to detect and characterize behavioral anomalies.

## Results

### Pipeline architecture

Inertial measurements of head kinematics were conducted on four Long-Evans rats placed inside a $70 \times 70$ cm arena. A custom multi-angle video capture system, operating synchronously, was used to independently document the animals' activity and compute 3D pose estimation (see Methods). The head-borne IMU contained a 3-axis accelerometer and a 3-axis gyroscope, providing head-referenced 3D vector data for acceleration (denoted as $\mathbf{a}$) and angular velocity (denoted as $\omega$), respectively. An estimation of the gravitational component of acceleration (denoted as $\mathbf{a_G}$) was computed using a previously validated method that combines $\mathbf{a}$ and $\omega$ [42] (see Methods), fully accounting for head orientation relative to gravity (hereafter referred to as head tilt). Subtracting $\mathbf{a_G}$ from $\mathbf{a}$ provided an estimate of the non-gravitational component of acceleration (denoted as $\mathbf{a_{nG}}$), representing acceleration generated by the animal's movements (Fig 1A, left).

The initial stage of DISSeCT aims to break down the multivariate time series composed of $\mathbf{a_G}$ and $\omega$ into non-overlapping segments with stationary statistics, in accordance with the notion that behavior can be conceptualized as a succession of elementary motifs [62,63]. To achieve this, we opted for kernel-based change-point detection (CPD) using a Gaussian Radial Basis Function kernel [64], implemented through an algorithm that ensures optimal segmentation with a linear computational cost [65,66] (see Methods). Unlike other techniques that focus solely on detecting shifts in mean values [67] or changes in variances [68], this CPD variant identifies changes in the entire distribution of the multivariate time series [69]. This allows for the detection of a broader range of segment types, effectively generalizing most traditional CPD methods. In our specific use case, this method outperformed a more traditional linear-kernel-based CPD approach in accurately isolating wet dog shake events (S3 and S5 Figs). The algorithm was calibrated to produce a median segment duration of 300 to 400 ms, aligning with the timescale of previously identified mouse behavioral modules [62]. Using this methodology, our dataset, totaling 8 h and 38 min of IMU recordings sampled at 300 Hz, was parsed into 44 322 segments (median duration: 377 ms, average duration: 701 ms, $q_{25}$ = 203 ms, $q_{75}$ = 820 ms) in about 5 min on a conventional workstation (see S4 Table).

The second stage of DISSeCT aims to perform cluster analysis of segments, treating each segment as an individual observation. It begins by extracting a catalog of 240 features that characterize segment properties. These features comprise multiple statistics on $\mathbf{a_G}$, $\mathbf{a_{nG}}$ and $\omega$, as well as time-frequency attributes of $\mathbf{a_{nG}}$ and $\omega$ along with directional information computed from $\mathbf{a_G}$ and $\omega$ (see Section 2 in S1 Text). One option, as explored in previous studies [13,15,16,18,70], would have been to implement the clustering step by partitioning a 2D representation of the feature space, although at the expense of losing potentially valuable information [71]. Instead, our approach involves applying a more conservative dimensionality reduction to maintain a high-dimensional space for clustering while achieving manageable runtimes. To accomplish this, standardized feature data were subjected to principal component analysis, retaining the first 30 principal components (PCs) as determined optimally by cross-validation (see Section 3 of S1 Text and S4A Fig). After an outlier removal step, a Gaussian mixture model (GMM) was then fitted to the data in the PC space. Unlike

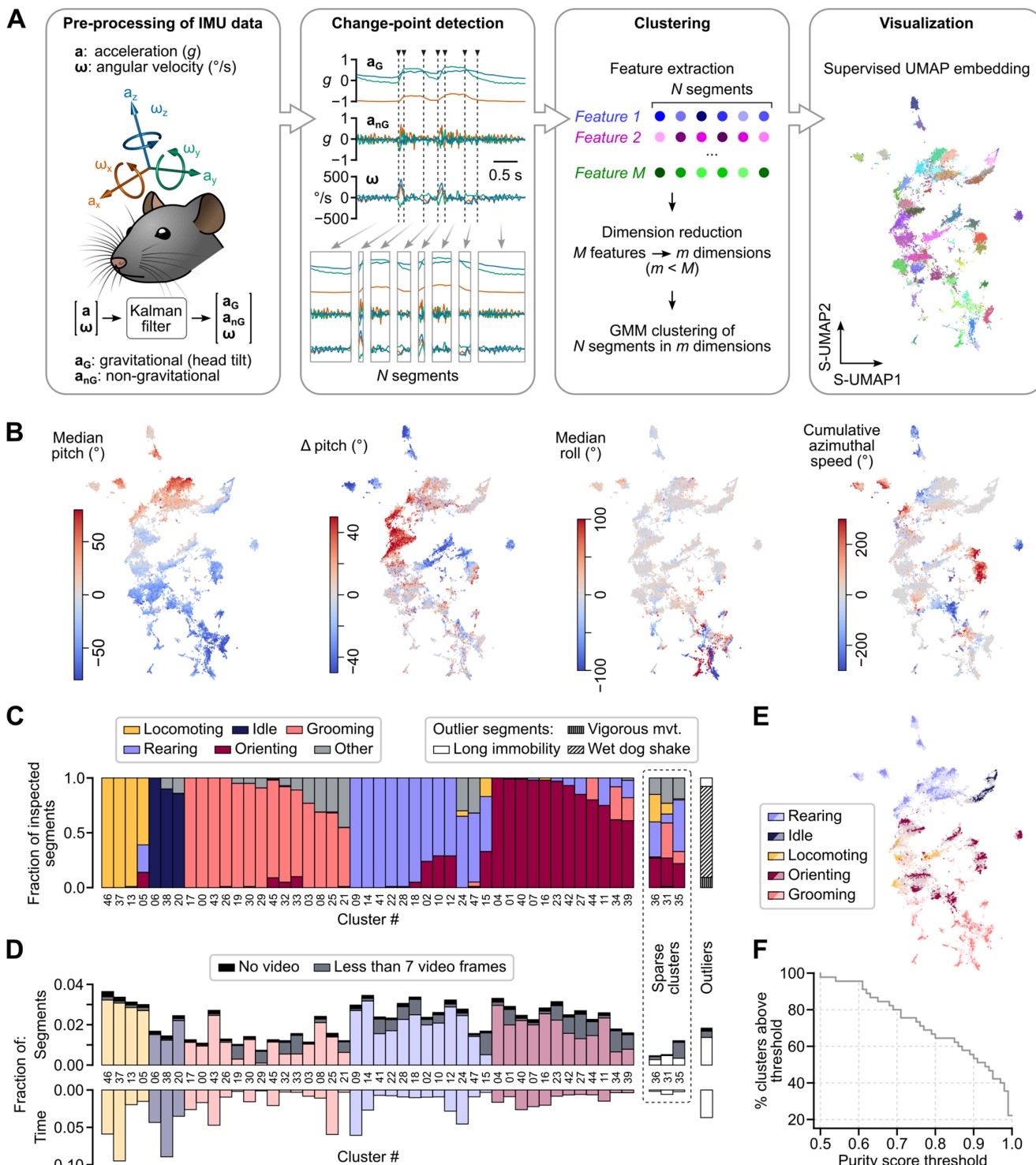

**Fig 1. Decomposition of rat behavior using DISSeCT.** (A) Pipeline architecture. (B) Example supervised UMAP segment embeddings with color-coded feature values. (C) Stacked bar plot depicting the assignment of main behavioral categories to the top 100 most representative segments within each cluster (see Methods). Clusters are grouped by their dominant category and sorted by decreasing purity score. In the case of outliers (rightmost bar), all segments were examined. (D) Top: fraction of the total number of segments represented by each cluster. For each cluster, the proportion of segments whose corresponding video snippet was either lacking or comprised less than 7 frames is highlighted in black and grey, respectively. Bottom: fraction of the total recording duration represented by each cluster (all segments included). (E) Supervised UMAP embedding showing segments color-coded according to their cluster's dominant main category (dark colors: inspected segments; light colors: other segments). (F) Fraction of clusters with a purity score above a given threshold, plotted as a function of that threshold.

standard clustering algorithms such as K-means [72], GMMs can capture a broader variety of cluster shapes—not just hyperspherical ones—by modeling the underlying data distribution as a mixture of Gaussian components [73]. Moreover, no assumptions are made regarding cluster sizes such that clusters with heterogeneous weights can be retrieved [74]. Finally, as a statistical model, GMM allows existing statistical tools to be employed. In particular, the Bayesian information criterion (BIC), which comes with theoretical backing, can be used to estimate the number of clusters present in the data [75]. Thus, considering a grid over the number of clusters and computing BIC for each GMM fit [76], we obtained 48 mixture components (S4B Fig). The corresponding 48 clusters were identified by assigning each segment to the mixture component to which it most likely belonged. We then applied semi-supervised UMAP (S-UMAP) to project segments into a 2D embedding of the standardized feature space, guided by the cluster identities obtained from the GMM. This embedding was not used for clustering, but to improve the readability of the GMM output by providing a clear visualization that preserves feature gradients (Figs 1B and S6A). Importantly, despite the supervision, the embedding retained meaningful information about cluster variance and inter-cluster relationships (S6B and S6C Fig).

To validate the usefulness of CPD, we conducted a post hoc comparison against a segmentation strategy based on fixed-length windows, a common default starting point in several behavioral analysis pipelines [63,77]. To this end, we divided IMU recordings into non-overlapping windows of 210 samples (700 ms), matching the average segment duration obtained with CPD, thereby preserving the total number of segments. We then applied the same downstream processing pipeline combining feature extraction, dimensionality reduction (PCA), and GMM clustering, retaining for these steps the hyperparameters obtained for varying-length segments. Specifically, we assessed how well GMMs, fitted to a PC representation of segments produced by each method, explained the underlying data. Fixed-length segmentation initially yielded higher log-likelihood values (S2E Fig), likely due to the subdivision of immobility periods into numerous low-variability windows that are easily modeled. This effect artificially inflates GMM performance compared to the CPD approach, which isolates immobility periods as single segments. After ablating periods of immobility prior to modeling (see Methods), the CPD-based strategy significantly outperformed the fixed-length approach (S2D and S2E Fig), highlighting the effectiveness of CPD in capturing meaningful structure for subsequent clustering.

In summary, DISSeCT provides a unified framework for pre-processing and parsing IMU data. It essentially relies on a fully unsupervised two-step process, wherein time series segmentation is used to isolate elementary kinematic motifs that are subsequently clustered in a high-dimensional space.

### Primary behavioral categories are parsed into distinct families of clusters

To assess the behavioral significance of IMU data decomposition using DISSeCT, we proceeded as follows: for each cluster, an observer was tasked with labeling the top one hundred segments with the highest posterior probabilities of belonging to it, and a *purity score* was computed to highlight the prevalence of the most frequent label (see Methods and S6 Table). With the exception of three sparse clusters, which represented highly scattered and underpopulated mixture components (see Methods and S7 Fig), all clusters exhibited a purity score of at least 0.5 (with two-thirds of them displaying scores above 0.8, Fig 1F) and could be clearly associated with a predominant behavioral category (orienting, rearing, grooming, locomoting or idle; see Fig 1C and 1D) occupying a distinct sub-region of the supervised UMAP embedding (Fig 1E). Purity scores tended to decrease when labeling was performed

on randomly-selected segments (S12C Fig), suggesting increased behavioral ambiguity in regions farther from mixture centroids and closer to cluster boundaries. A significant fraction of clusters (19 out of the 45 non-sparse clusters), however, displayed similar purity scores irrespective of the segment selection method (S12E Fig). This analysis shows that the decomposition of IMU data obtained with our pipeline largely aligns with the major classes of rodent activities. The fact that each behavioral category was subdivided into several clusters, with three of them ("orienting", "rearing", and "grooming") comprising no fewer than 12 clusters each (Fig 1C), indicated a higher level of granularity where each category is further broken down into multiple components. This prompted us to conduct a comprehensive examination of the GMM output in order to assess the level of detail achieved with DISSeCT.

## Identification of clusters mapping the repertoire of rapid changes in head orientation

Considering that our primary data included direct measurements of 3D head angular velocity, we anticipated that DISSeCT would be able to discern motor patterns associated with rapid changes in head orientation. A preliminary examination of the output of the CPD step indeed suggested that such movements were isolated as individual segments (see S1 Movie). Among the features extracted from all segments, we focused on net angular speed, calculated by dividing the net head angular displacement (from segment start to finish) by the segment's duration. The distribution of per-cluster average net angular speed values exhibited a bimodal pattern, with a cutoff of about 150°/s (Fig 2A). The upper half of this distribution contained 18 clusters representing brief ($179 \pm 45$ ms) head rotations associated with the "orienting", "rearing", and "grooming" categories (Fig 2A and 2B), and accounting for 37% of all segments (11% of total recording duration). With the exception of the rearing termination phase, all these clusters corresponded to rotations of 30–50° (mean: $42.8 \pm 8.8°$, Fig 2B), regardless of the net angular speed ($r = 0.033, p = 0.90$). This likely reflects the fact that the average segment duration decreased as a function of the average net angular speed ($r = -0.86, p = 1.6 \times 10^{-5}$), suggesting that rapid head movements might be tuned to produce relatively invariant head angular displacement.

Behavioral categories could be distinguished based on the head pitch angle and net azimuthal speed (Fig 2C). Grooming-associated rotations, corresponding to single strokes or brief head repositioning between grooming bouts (S1 Movie), occurred with the head pitched downward ($< -45°$) and showed limited change in azimuth. Rearing-associated rotations, mainly corresponding to rearing initiation or termination, took place with the head pitched upward ($> 0°$). "Orienting" movements predominantly fell within an intermediate range of pitch angles, accompanied by either minimal or significant changes in azimuth.

These observations show that our pipeline offers an effective method for isolating and sorting rapid head reorientations in 3D. A visually informative overview depicting the corresponding repertoire of such movements was obtained using custom 2D projected maps showing the segment-wise spherical trajectories of $\mathbf{a_G}$ in the head reference frame and of the animal's heading in an absolute (external) reference frame (see Section 2.2 in S1 Text and S2 Movie). These maps are shown together with the associated 3D poses and IMU signals in Fig 2D.

We noticed that some clusters contained segments belonging to different categories (Fig 2C), interpreting this as a consequence of distinct behavioral motifs sometimes generating similar head movements. For example, rearing initiation and upward head rotation may share common head kinematic features, differing primarily in whether the forepaws are leaving the ground or not. This point is further investigated in a later section where we

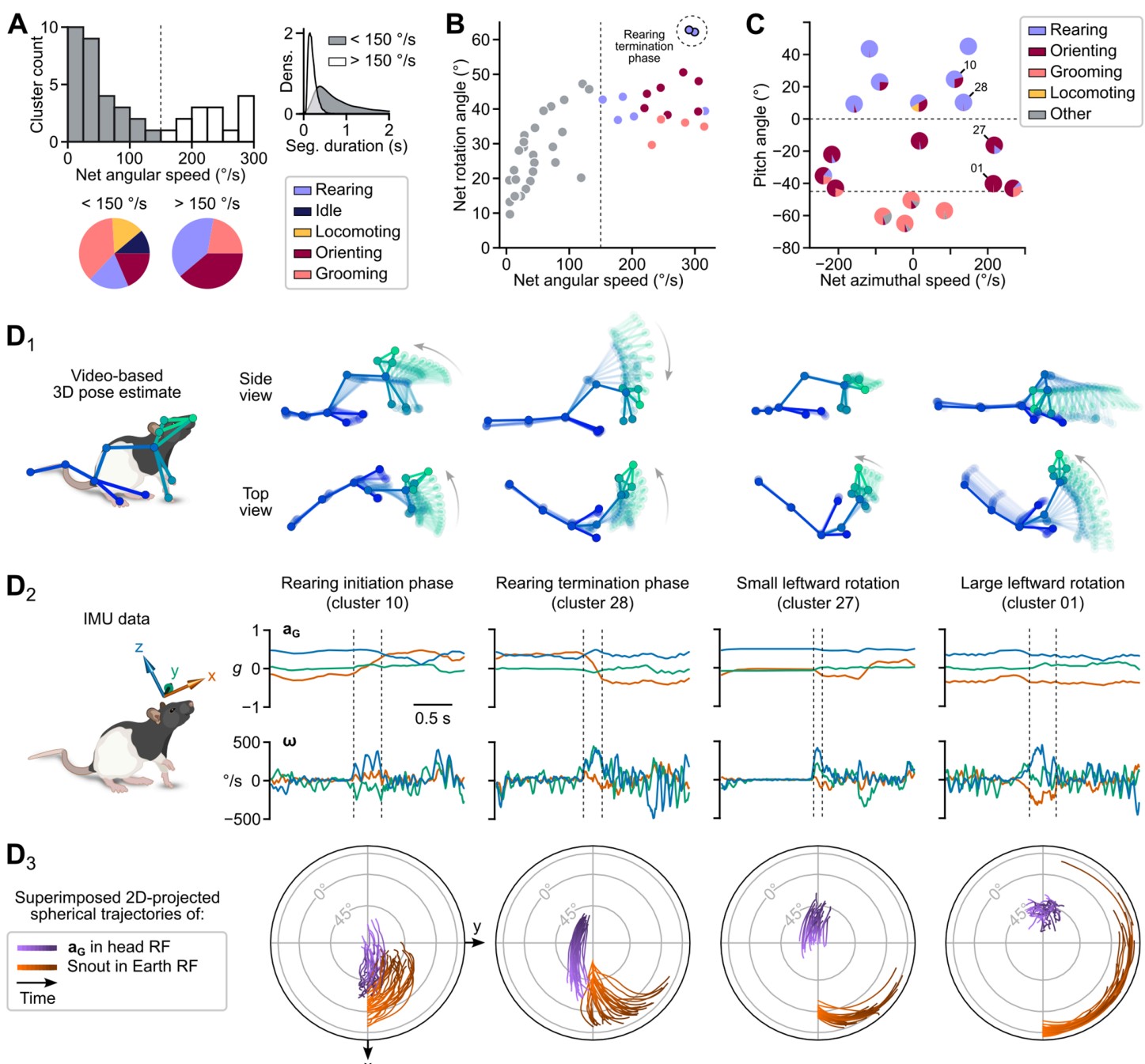

**Fig 2. Diversity and fine dynamics of rapid changes in head orientation**. (A) Distribution of net angular speed values (median of per-segment average values) for all clusters. The dashed vertical line at 150°/s corresponds to the threshold for selecting clusters representing rapid head rotations (highlighted in white in the histogram). Pie charts illustrate the proportion of cluster-associated behavioral categories (one unique category per cluster) for each side of the distribution. The upper right inset shows a density estimate of segment duration for each side of the distribution. (B) Net rotation angle plotted against net angular speed (median of per-segment average values) for each cluster. Clusters are colored according to their associated behavioral category (using the same color code as in A and Fig 1C). (C) Head pitch angle (see Methods) plotted as a function of net azimuthal speed (median of per-segment average values) for each cluster. Each cluster is displayed as a pie chart showing the behavioral categories attributed to the top 100 segments of the cluster (cf. Fig 1C). (D1–3), Detailed examination of four example clusters (identified in C). Each column corresponds to one cluster. D1: Skeletons representing the animal's 3D pose dynamics during one example segment. Zero transparency corresponds to the last video frame of the segment. D2: Gravitational acceleration and angular velocity signals ($a_G$ and $\omega$, respectively) for the same segment. Dashed lines represent segment boundaries. D3: Joint visualization of the trajectories of $a_G$ in the head reference frame (purple) and of the animal's heading in the Earth reference frame (orange) for the top 30 segments of each cluster (see subsection 2.2 in S1 Text; RF: reference frame).

show that such ambiguities can be resolved by taking into account the sequences of behavioral motifs.

## Locomotion splits into different regimes, including a mobile olfactory exploration module

As shown in Fig 1C and 1D, the "locomotion" and "idle" categories, which accounted for 18% of all segments and 36% of the total recording duration, contained fewer clusters compared to other behavioral categories, suggesting a higher degree of kinematic homogeneity. When considering actual animal displacement, three of the "locomotion" clusters emerged as those associated with the highest speed (Fig 3A), representing 68% of the time when the animal's speed exceeded 15°/s. Two of the "idle" clusters emerged as those with the maximal fraction of time spent immobile (defined using a head angular speed threshold; see Methods and Fig 3A). One of these (cluster 6) corresponded to true periods of immobility, while the other (cluster 38) contained longer duration segments during which the animal was nearly immobile but exhibited small head movements (Fig 3A and 3B).

To understand what caused the clustering algorithm to isolate locomotor activity, we examined the head kinematic features associated with the three "locomotion" clusters highlighted in Fig 3A. Two clusters (46 and 13), corresponding to second-scale fast locomotion bouts, were notably characterized by sawtooth-like patterns in the $z$ component of non-gravitational acceleration (Fig 3B and 3C) synced with each step cycle, as previously described from head acceleration measurements in running rats [28]. This signature likely reflects a bobbing head motion associated with the successive cycles of storage and release of kinetic and potential energy in the animal's skeletal muscles [78]. Continuous wavelet transform revealed that this oscillatory regime took place in the 5–7 Hz band, consistent with the known step cycle duration during trotting in rats [79].

The third cluster (37) corresponded to a strikingly different locomotor regime in which most of the power was concentrated around 10 Hz in the $x$ and $y$ axes of non-gravitational acceleration and angular velocity, respectively (Fig 3D). This signature, reflecting coordinated fore-aft motion and pitch rotation, has already been described as linked with sniffing in rodents [31,80]. Compared to the other two "locomotion" clusters, this locomotor regime was also characterized by a larger downward head pitch angle of about 40° (Fig 3E). Video inspection clearly showed that this third locomotion cluster corresponded to episodes of olfactory exploration, during which animals slowly moved forward maintaining their snout close to the floor (S3 Movie).

Head pitch was not the only feature distinguishing mobile olfactory exploration from rapid locomotion. While the distribution of head roll was unimodal in the first case, it appeared to be bimodal in the second (Fig 3F), peaking at 5–10° toward the left and right sides. This phenomenon did not result from variability in IMU placement on the head, as segments from the same animal, and even from the same session, showed both orientations. During rapid locomotion, per-segment average roll angles showed a weak correlation with net azimuthal speed ($r = 0.15$, $p < 1 \times 10^{-4}$). This suggests that head roll tends to occur when the animal follows a curved path, with the head tilting toward the concave side of the trajectory.

## The vigor of olfactory exploration can be tracked by analyzing its spectral signature

Having observed a head kinematic signature linked with sniffing during slow locomotion, we asked whether a similar pattern could be detected in other behavioral contexts. Visual inspection of segments and their associated video snippets indeed suggested that $\approx 10\,\text{Hz}$

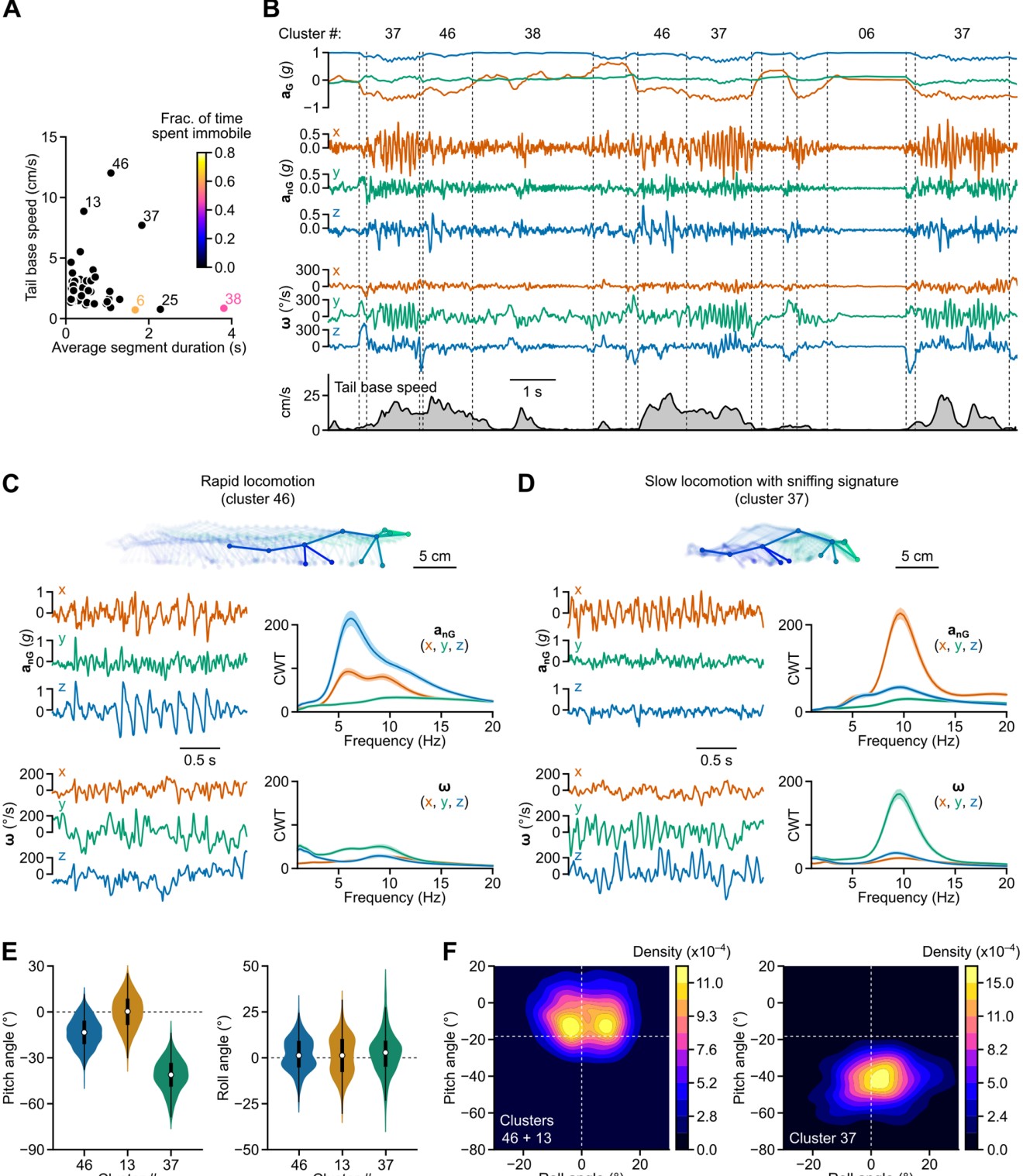

**Fig 3. Identification of two locomotor regimes.** (A) Tail base speed (median of per-segment average values) plotted against average segment duration for each cluster. The color scale represents the median fraction of time spent immobile, computed for each segment based on head angular speed (see Methods). Cluster numbers are indicated for three "locomotion" (13, 37, and 46) and two "idle" clusters (6 and 38). (B) Upper panels: Example traces of head gravitational and non-gravitational acceleration and angular velocity ($a_G$, $a_{nG}$ and $\omega$, respectively). Lower panel: Concurrent horizontal tail base speed computed from 3D pose estimates. Dashed vertical lines indicate segment boundaries. Corresponding cluster numbers are shown above the traces. (C) Top: Skeletons representing the animal's 3D pose dynamics during 30 video frames (1 s) extracted from the segment with the highest posterior

probability of belonging to cluster 46 (associated with rapid locomotion). Zero transparency corresponds to the last frame. Left: Example traces of head non-gravitational acceleration and angular velocity ($\mathbf{a_{nG}}$ and $\omega$, respectively) for the same segment. Right: Continuous wavelet transform (CWT) coefficients for all three axes of $\mathbf{a_{nG}}$ and $\omega$. Solid lines represent the average of per-segment median log-magnitude values (shared area: 95% confidence interval). (D) Similar to panel C, but for cluster 37 (corresponding to mobile olfactory exploration). (E) Distribution of per-segment average head pitch and roll angles for the three "locomotion" clusters identified in A. The data underlying this panel can be found in S1 Data. (F) 2D histograms showing the distribution of per-segment average head pitch and roll angles for clusters associated with rapid locomotion (left) and mobile olfactory exploration (right). Dashed lines indicate the resting pitch and roll angles, computed as the median of per-segment average pitch and roll angles for all segments.

rhythmicity in the fore-aft non-gravitational acceleration and pitch angular velocity was a prevalent feature of head IMU recordings also manifesting in other clusters. To identify these clusters, we devised a "sniffing score" reflecting the magnitude of this spectral signature based on a per-segment ratio of continuous wavelet transform coefficients in specific frequency bands (see Fig 4A and Methods).

In addition to cluster 37, associated with the slow locomotor regime identified in Fig 3D, this approach revealed six clusters exhibiting a high sniffing score (Fig 4B) and associated with the "orienting" and "rearing" categories. The strongest scores were observed for three "orienting" clusters not part of the rapid head movements identified in Fig 2B, corresponding to small upward pitch movements or rightward or leftward head rotations with the head pitched down (Fig 4C), and displaying net angular speed values in the 35–60 °/s range. The two clusters with lower scores corresponded to rearing initiation and high rear with a vertically stretched body (Fig 4C). These observations, along with the bimodal distribution of average head pitch angles associated with these clusters (Fig 4C), suggest that animals preferentially engaged in robust sniffing either when investigating the floor or during rearing.

The average peak frequencies observed for these eight clusters with high sniffing score ($9.1 \pm 2.1$ Hz and $8.2 \pm 1.7$ Hz for fore-aft non-gravitational acceleration and pitch angular velocity, respectively) were in the upper range of sniffing rates reported for rats during odor discrimination in stationary or head fixed conditions (6–9 Hz) [31,81], and were consistent with those observed in freely-moving animals tracking an odor trail (8–12 Hz) [82]. Overall, these results indicate that head IMU signals can be leveraged to obtain a detailed and quantitative readout of olfactory exploration using a simple metric that can be applied independently of our pipeline.

## The main types of grooming activities distribute into separate clusters with distinctive spectral signatures

Self-grooming is a major component of the rodent behavioral repertoire, encompassing a variety of actions often performed sequentially [83]. Monitoring grooming activities through video recordings is however challenging due to the animal's contorted posture (e.g., during body licking) or the rapid movements involved (e.g., during scratching). To assess DISSeCT's ability to parse this family of actions, we conducted a detailed examination of the 14 clusters assigned to the "grooming" category during segment inspection (Fig 1C), which accounted for 19% of all segments and 23% of the total recording duration.

These clusters were characterized by a specific range of head tilts, with the less ambiguous ones (corresponding to body licking) exhibiting a unique combination of head pitch and roll angles (see cluster 17 in Fig 5A). Unexpectedly, their most distinctive feature was a clear deviation from a linear relationship observed for all other clusters between the correlation of $x$ and $z$-axes gyroscope measurements and the head pitch angle ($r = 0.95$, $p = 2.2 \times 10^{-16}$; Fig 5A). In other words, the coupling between roll and yaw head rotations during grooming appears to depart from a general biomechanical rule linking it with head pitch in all other conditions.

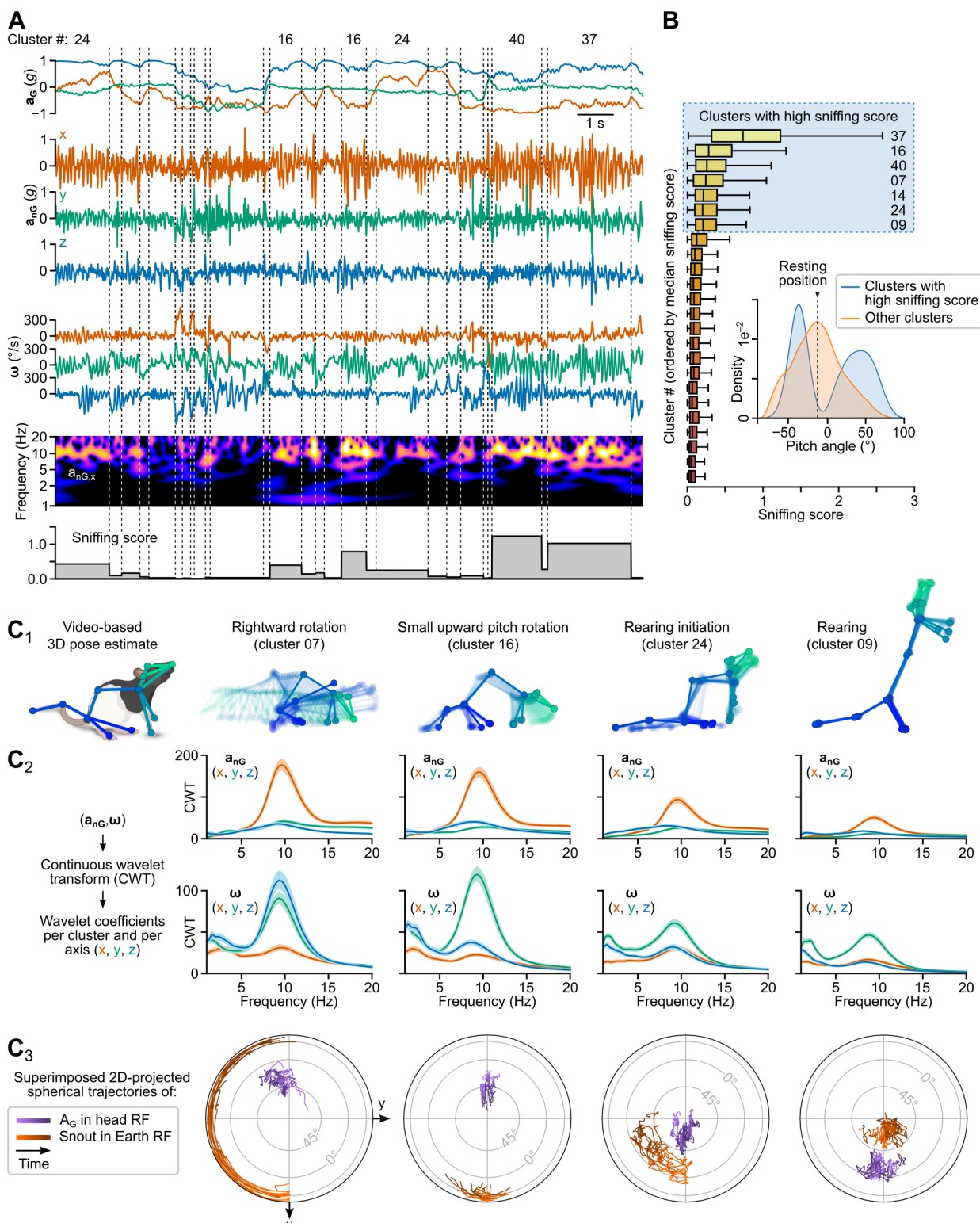

**Fig 4. Identification of a sniffing signature.** (A) Upper panels: example traces of head gravitational acceleration ($\mathbf{a_G}$), non-gravitational acceleration ($\mathbf{a_{nG}}$), and angular velocity ($\boldsymbol{\omega}$). Lower panels: log-magnitude of continuous wavelet transform coefficients computed for fore-aft non-gravitational acceleration ($x$-axis component of $\mathbf{a_G}$), and per-segment sniffing score values. Dashed vertical lines indicate segment boundaries. Corresponding cluster numbers are indicated above the traces. (B) Box plot of sniffing score values shown for 24 clusters. Clusters are sorted by decreasing median sniffing score. Inset: Distribution of per-segment median head pitch angles computed for the seven clusters with

highest sniffing scores (blue curve) and for all other clusters (orange curve). $C_{1-3}$, Detailed examination of four example clusters with high sniffing scores. Each column corresponds to one cluster. $C_1$: Skeletons representing the animal's 3D pose dynamics during one example segment. Zero transparency corresponds to the last video frame of the segment. $C_2$: Log-magnitude of continuous wavelet transform (CWT) coefficients (average of per-segment median values) for all three axes of $a_{nG}$ and $\omega$, shown for each example cluster (shaded area: 95% confidence interval). $C_3$: Joint visualization of the trajectories of $a_G$ in the head reference frame (purple) and of the animal's heading (orange) for the top 10 segments of each cluster (see subsection 2.2 in S1 Text; RF: reference frame).

To assess whether "grooming" clusters corresponded to ethologically distinct actions, we relied on the labels assigned during the inspection phase (Fig 5B; see Methods). The less ambiguous clusters (where more than 85% of inspected segments received the same label) corresponded to body licking (clusters 0, 17, and 26), facial grooming (cluster 43), head and neck scratching (cluster 45), and hindpaw licking (cluster 30), each exhibiting a characteristic spectral signature (Fig 5C). Another group of clusters had less homogeneous labels, with short bouts of facial grooming mixed with non-grooming related head orienting movements (cluster 8), or individual body licking strokes mixed with brief head movements occurring when the animal investigated the corners of the arena with a tilted head posture (cluster 21). The non-grooming-related segments included in these clusters likely correspond to head motion patterns whose kinematics resemble those encountered during grooming. Kinematic ambiguities may also explain why ethologically distinct grooming actions were mixed in the same clusters (see clusters 43 and 25, Fig 5B). Finally, a last group of clusters (clusters 19, 29, 32, and 33) contained various head orienting movements executed in the context of grooming episodes (at episode onset or offset or between adjacent grooming bouts).

These results show that our pipeline successfully isolated the main components of grooming as a family of clusters with distinctive postural, kinematic, and spectral signatures. They also point out kinematic ambiguities which could be taken into account to design strategies aiming at achieving an even greater separation of grooming actions.

## Behavioral patterns can be unveiled from cluster sequences

As detailed in the previous sections, DISSeCT effectively parsed head IMU signals into various behaviorally relevant components. To assess how these components assemble over time and potentially form sequential patterns, we fitted a simple hidden Markov model (HMM) with categorical emissions to the cluster sequence (i.e., where categories correspond to clusters). Hyperparameter optimization resulted in 16 states, each associated with a vector of emission probabilities for the different clusters (see Methods and Fig 6A). We observed that states often appropriately gathered clusters corresponding to the same movement executed in opposite leftward and rightward directions (S8 Fig). Additionally, the highest emission probabilities for a given state most often corresponded to clusters belonging to the same behavioral category, allowing each state to be linked to a specific category (Fig 6A).

The state transition matrix revealed that the most likely transitions occurred within the "rearing", "grooming", and "orienting" categories (Fig 6B), indicating that these behaviors tended to manifest as multi-state sequences rather than isolated events. States associated with the "rearing" and "grooming" categories, in particular, almost never transitioned to one another, while the "locomoting", "rearing", and "orienting" categories did exhibit specific temporal relationships. These observations were further confirmed by examining the cluster transition matrix (S9 Fig).

Characteristic intra-category sequences included back-and-forth switching between states 1 and 3, representing an alternation between head scratching or hindpaw licking episodes and brief head orienting movements (Fig 6C and $6D_1$), and a 3-state chain (states 5, 8, and

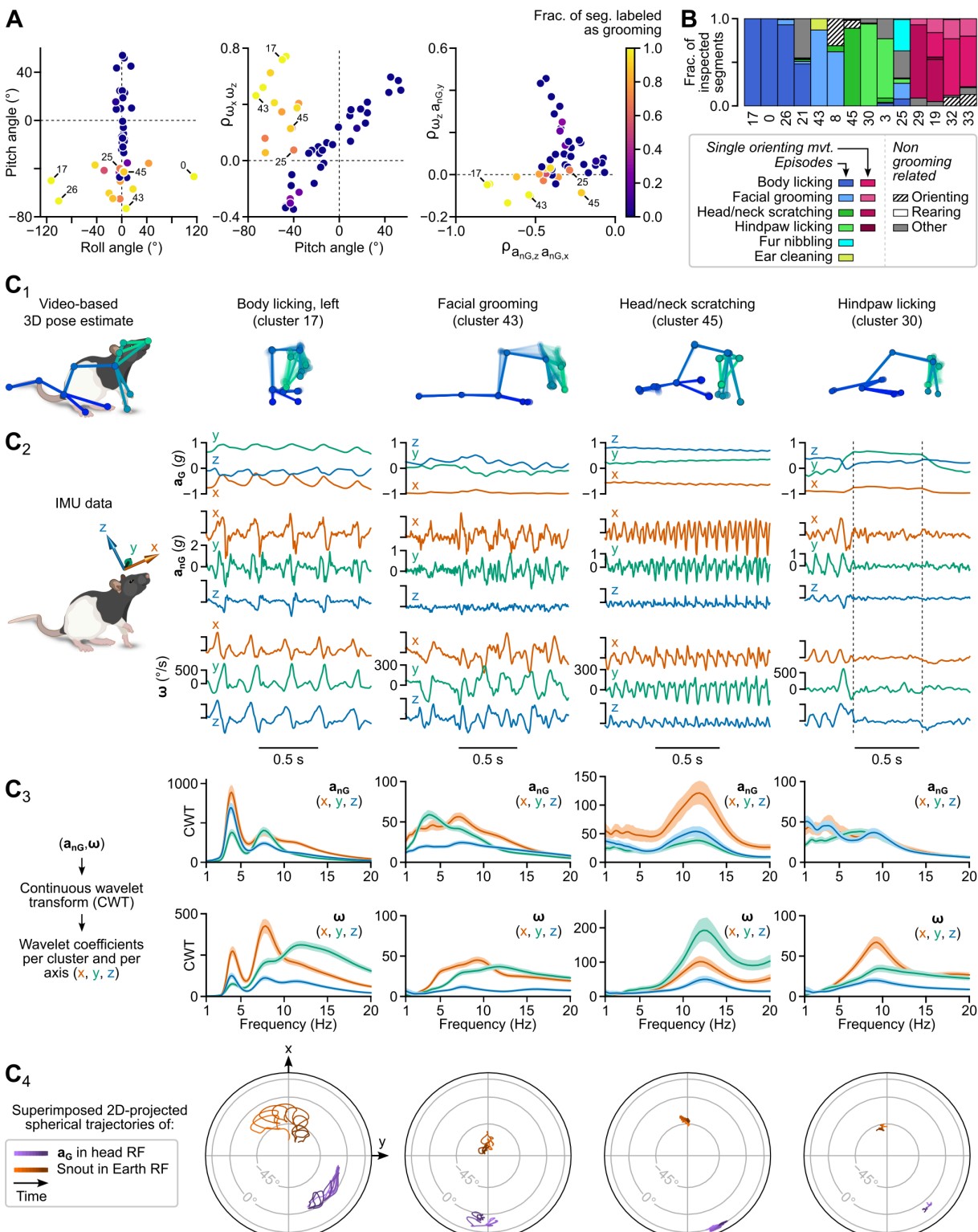

**Fig 5. Parsing of grooming components.** (A) Distinctive features of grooming-related clusters. Left: head pitch vs. roll angles. Middle: correlation between angular velocities about the $x$ and $z$ axes ($\omega_x$ and $\omega_z$, respectively) plotted against head pitch angle. Right: correlation between angular velocity about the $z$ axis ($\omega_z$) and medio-lateral non-gravitational acceleration ($a_{nG,y}$) plotted against the correlation between the head-vertical and fore-aft components of non-gravitational acceleration ($a_{nG,z}$ and $a_{nG,x}$, respectively). These panels show per-cluster median values of per-segment average values. The color scale indicates the fraction of segments identified as belonging to a grooming sequence during inspection of the top 100 segments of each cluster (see Method). (B) Stacked bar plot depicting the assignment of labels to inspected segments

within each grooming-related cluster. $C_{1-3}$, Detailed examination of four example clusters. Each column corresponds to one cluster. $C_1$: Skeletons representing the animal's 3D pose dynamics during one example segment. Zero transparency corresponds to the last video frame of the segment. $C_2$: Gravitational and non-gravitational acceleration ($\mathbf{a_G}$ and $\mathbf{a_{nG}}$, respectively), and angular velocity ($\omega$) for the same segment. Average segment-wise median log-magnitude of continuous wavelet transform (CWT) coefficients for all three axes of $\mathbf{a_{nG}}$ and $\omega$ (shared area: 95% confidence interval).

10) representing rearing episodes (Fig 6C). Prominent inter-category sequences included an alternation between rearing episodes and locomotion bouts (Figs 6C, $6D_2$, and S10A), and a 3-state sequence (states 6, 2, and 7) representing episodes during which the animal lifted its head up while keeping its front paws on or close to the ground (Figs 6C, $6D_3$, and S10B).

One key property of HMMs is that the probability of being in a given state for a particular observation changes over time depending on the local context. As a result, segments assigned to a given cluster by the GMM but occurring in different behavioral contexts might be associated with different states, potentially reflecting different behavioral motifs. To examine this, we plotted posterior probabilities for each cluster (S11A Fig) and found clusters linked to multiple states. A detailed analysis of two dual-state clusters revealed that they could indeed be split into kinematically similar but ethologically distinct head movements based on the posterior probability of each segment. Cluster 12 was parsed into rearing segments (linked with state 8) and segments during which the head was pitched-up but the front paws remained on or near the ground (linked with state 2), as shown in S11B Fig. Cluster 19 was parsed into brief orienting movements executed between head scratching and hindpaw licking bouts (linked with state 3) and single strokes during facial grooming or body licking (linked with state 11), as shown in S11C Fig.

These observations demonstrate that a simple categorical HMM can be effectively used as a curation step after DISSeCT to merge ethologically-similar clusters and resolve certain kinematic ambiguities by splitting clusters containing similar head movements performed in different contexts.

## Comparison with video-based behavioral analysis

In our rat dataset, IMU recordings were acquired concurrently with multi-view video recordings (see Methods), enabling direct comparison of our pipeline with 3D pose-based behavioral mapping tools. To analyze 3D pose data, we used Keypoint-MoSeq (KpMS), an unsupervised pipeline that applies a hierarchical graphical model to jointly perform pose estimate refinement, segmentation, and clustering, and which has been explicitly validated for 3D pose data [17]. KpMS parses a low-dimensional representation of keypoint trajectories into behavioral syllables, each composed of multiple instances. Functionally, these syllables and instances are equivalent to the clusters and segments identified by DISSeCT.

KpMS was parameterized to match the median duration of syllable instances to that of DISSeCT segments and previous studies [17,62] (see Methods). Under these conditions, it produced a greater proportion of short (< 0.2 s) and intermediate (0.5–2 s) instances (S12A Fig) and identified fewer syllables (S12B Fig). In-depth analysis was conducted on syllables accounting for more than 5% of the total recording duration. Their behavioral content was evaluated via visual inspection, following the same protocol used for DISSeCT clusters (see Methods). A clear behavioral category could be assigned to nearly all KpMS syllables. As expected, purity scores decreased following randomization, though not uniformly across clusters and syllables (S12C to S12E Fig), as well as across behavioral categories (S12G Fig). Syllable-wise purity scores were less affected than cluster-wise scores (S12F Fig). When purity

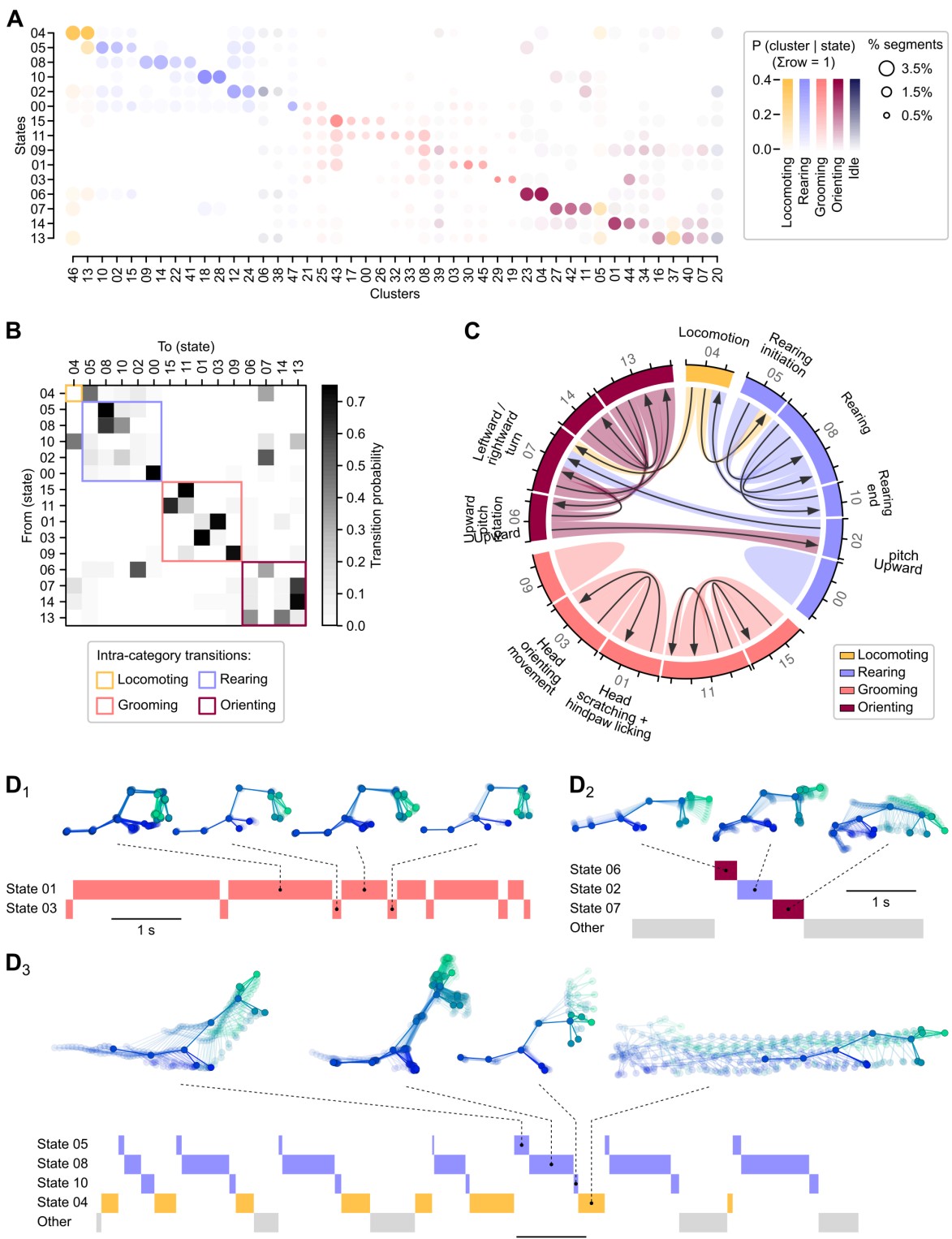

**Fig 6. Identification of behavioral sequences using a post-hoc categorical hidden Markov model.** (A) Cluster emission probabilities (columns) for each hidden state (rows). The dots in each column are colored according to the cluster's associated main behavioral category as defined in Fig 1C, with the opacity representing probability values and dot area representing how much the cluster contributed to the total number of segments. Note that state 12, which contains outlier segments and sparse clusters (see Methods and S7 Fig) is not represented in this plot. (B) Hidden state transition matrix. States are ordered according to their dominant main behavioral category. Colored squares highlight intra-category transitions. (C) Chord plot representing transition probabilities between states. Arrow

width represents transition probability (tick spacing: 0.5). States are colored according to their dominant main behavioral category. Only transitions probabilities greater than 0.25 are represented. **D$_{1-3}$**, Representative ethograms for three types of sequences. D$_1$: Alternation between head scratching or hindpaw licking episodes (state 1) and brief head orienting movements (state 3). D$_2$: Single episode during which the animal is lifting its head up before executing a turn. D$_3$: Alternation between rearing episodes (states 5, 8, and 10) and locomotion bouts (state 4). For each ethogram, skeletons representing the animal's 3D pose dynamics are shown for example segments. Zero transparency corresponds to the last video frame of the segment.

scores were computed at the level of the main behavioral categories, this difference was more nuanced, with DISSeCT achieving comparable or higher scores than KpMS for the "idle", "grooming", and "orienting" categories (S12G Fig). DISSeCT subdivided the "idle", "grooming", "rearing", and "orienting" categories into more clusters, whereas KpMS identified a greater number of "locomotion" syllables. This difference likely reflects the complementarity of both methods: DISSeCT focuses on fine-grained head kinematics, while KpMS—relying on whole-body pose—distinguishes among multiple locomotion bouts based on trajectory information (see S4 and S5 Movies).

To quantify the alignment between the two approaches, we computed contingency matrices (S13 Fig). DISSeCT and KpMS showed broad agreement on the "idle", "grooming", and "rearing" categories, but less consensus on "locomoting" and "orienting" (S13B and S13D Fig). This may be partly explained by the fact that some of DISSeCT's "orienting" segments are embedded within KpMS's curved locomotion bouts, as well as by the inherent ambiguity—apparent to human observers—of short video snippets that combine orienting movements with locomotor activity.

Notably, several behavioral motifs captured by DISSeCT were missed by KpMS. For example, while KpMS successfully differentiated body licking from facial grooming, it failed to isolate scratching and hindpaw licking as distinct syllables (compare Figs 5B and S12H). KpMS also did not isolate wet dog shake events, which composed the vast majority of DISSeCT's outlier segments (Fig 1C). Part of these discrepancies likely stem from the higher temporal resolution required to resolve such rapid behaviors, compounded here by the tenfold difference in sampling rates between IMU and video data (300 Hz vs. 30 Hz).

Differences between the outputs of DISSeCT and KpMS likely stem from both the nature of their input data and their underlying mathematical frameworks: DISSeCT yields a finer decomposition of behaviors with a marked head kinematic signature, whereas KpMS offers a coarser clustering—except for locomotion—based on whole-body dynamics. Quantitatively, the Adjusted Rand Index comparing the two clusterings was 0.19 (excluding DISSeCT's outlier segments), indicating limited structural overlap.

The pipelines also differ substantially in computational demands. Despite the higher IMU sampling rate, DISSeCT—which is CPU-based—ran five times faster than the GPU-based version of KpMS (see S4 and S5 Tables), not including the additional cost of video-based 3D pose estimation.

## Unsupervised analysis of behavioral alterations in a mouse model of Parkinson's disease and levodopa-induced dyskinesia

One advantageous feature of GMMs is their probabilistic nature, which makes them suitable for estimating how well a new dataset conforms to a previously-established model and thus for detecting anomalies. To test the applicability of DISSeCT for quantifying behavioral changes, we used a classical toxin-induced mouse model of Parkinson's disease (PD). 6-OHDA-lesioned mice (PD mice, $n = 12$) were obtained through unilateral nigrostriatal dopaminergic denervation using 6-OHDA injection (see Methods) while a control group

received saline injection (sham mice, $n = 4$). PD mice exhibited decreased locomotion, impaired movement initiation and increased ipsiversive rotations relative to the lesion side, as previously described [84]. PD mice were then treated with daily injections of levodopa (L-DOPA), the first-line drug for the management of PD motor symptoms, along with sham animals. L-DOPA induced abnormal involuntary movements (AIMs) in PD mice, mimicking the common complication of this treatment in PD patients, known as levodopa-induced dyskinesia (LID). These AIMs, which affect the musculature on the side contralateral to the lesion [84], were accompanied by contraversive rotations.

Weekly IMU recordings were performed on PD and sham mice before and during L-DOPA treatment (Fig 7A). DISSeCT was then run on sham data to generate a GMM representing the behavior of healthy animals (sham-GMM, see Methods). A joint S-UMAP embedding, computed across all animals in all conditions using cluster labels from the sham-GMM, revealed a similar global structure for sham and PD mice, while entirely new regions emerged for PD mice during the LID phase (Fig 7B). These novel regions were also clearly visible in fully unsupervised embeddings based solely on kinematic features (S14 Fig), confirming the emergence of L-DOPA-specific behavioral patterns that are not present in healthy or untreated PD mice.

To quantify how much the motor repertoire of PD mice deviated from that of healthy animals, we calculated the average log-likelihood of segments under the sham-GMM. To mitigate potential overfitting, the score for each sham animal was computed using a leave-one-out approach (see Methods). This "score under sham model" (SUSM) was only slightly lower in PD mice compared to sham mice before L-DOPA administration (Fig 7C). After L-DOPA injection, the SUSM of PD mice decreased substantially (Fig 7C), indicating that the sham-GMM poorly explained their behavior during the LID phase. Given the characteristic rotational behavior in this model and its inversion during the LID phase, we quantified rotations using IMU data as previously described [42]. PD mice displayed mild but significant ipsiversive rotations relative to the lesion side ($-1.73 \pm 0.80$ turns per minute, vs. $-0.12 \pm 0.10$ for sham mice; $p = 1.1 \times 10^{-3}$, Wilcoxon test) before L-DOPA injection and a pronounced circling behavior in the opposite direction after injection ($17 \pm 9$ turns per minute). Rotations during the LID phase negatively correlated with the SUSM ($r = -0.89$, $p = 9.0 \times 10^{-5}$; Fig 7D), suggesting that most of the L-DOPA-induced behavioral remapping captured by our pipeline is linked with an overrepresentation of contraversive movements. Notably, these rotations also negatively correlated with the SUSM measured before injection ($r = -0.71$, $p = 9.5 \times 10^{-3}$; Fig 7E), suggesting that changes occurring in the PD state are predictive of the overall response to L-DOPA.

To further characterize behavioral changes in PD mice, we examined how frequently sham-GMM clusters were used in these mice in comparison with sham mice (Fig 8A). While the S-UMAP embedding appears similar between sham and PD mice before L-DOPA, this cluster usage analysis revealed clear behavioral differences. Before L-DOPA injections, PD mice predominantly employed two clusters: one corresponding to episodes of immobility and the other to slow, mobile olfactory exploration with ipsiversive turns. During the LID phase, cluster usage shifted to a series of five clusters, all associated with contraversive movements (Fig 8B and 8C). Among clusters whose usage did not change drastically, we focused on one representing upward pitch movements, which exhibited a slight positional shift in the embedding space across groups and conditions (Fig 8D). This shift occurred along a gradient of head roll angles, with PD mice segments concentrated on opposite sides of the gradient before and after L-DOPA injection. This rearrangement corresponded to a mild ipsiversive bias of upward pitch movements in PD mice, which reversed to the opposite side under

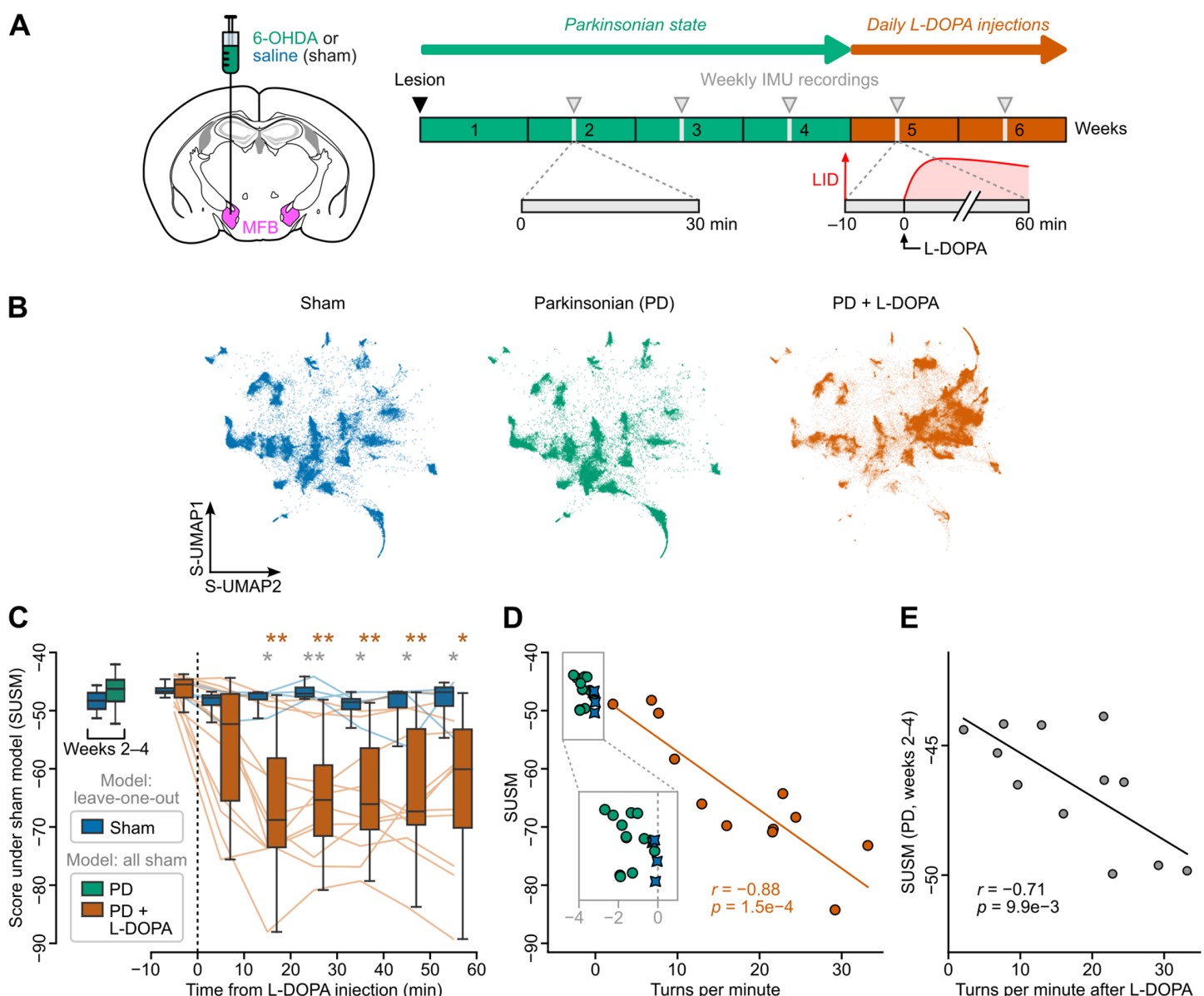

**Fig 7. Macroscopic analysis of motor impairments in a mouse model of Parkinson's disease and levodopa-induced dyskinesia.** (A) Experimental protocol. Left: drawing of a coronal brain slice (1.2 mm posterior to Bregma, adapted from [111]) showing the unilateral injection of 6-OHDA (PD mice, $n$ = 12) or saline (sham mice, $n$ = 4) into the medial forebrain bundle (MFB, purple). Right: protocol timeline (see Methods for details). The six-week protocol began with 6-OHDA (PD mice) or saline (sham mice) injection and IMU implantation. Weekly IMU recordings (gray arrowheads) were performed during weeks 2 to 4. Daily levodopa (L-DOPA) injections started in week 5, with IMU recording sessions conducted during weeks 5 and 6. Recordings began 10 min before injection. Each L-DOPA injection triggered an episode of abnormal involuntary movements (LID: L-DOPA induced dyskinesia, symbolized by a red curve). (B) S-UMAP embeddings for sham mice (left, blue) and for PD mice before (middle, green) and during (right, orange) L-DOPA treatment, using a clustering model fitted on sham mice data. (C) Left: box plots of the score under sham model (SUSM) for sham ($n$ = 4) vs. PD ($n$ = 12) mice before L-DOPA treatment (weeks 2 to 4). Right: box plots of the SUSM computed for the same mice over 10-min bins right before and after L-DOPA injection (weeks 5 and 6). Lines show the time course of the SUSM for individual PD (orange) and sham (blue) mice. Gray asterisks denote significant differences between PD vs. sham mice SUSM values for the same time bins (unpaired pairwise t-test with Bonferroni correction; *: p-value < 0.05; **: p-value < 0.01). Orange asterisks denote significant differences between PD mice SUSM values before vs. after L-DOPA injection (paired pairwise t-test with Bonferroni correction; *: p-value < 0.05; **: p-value < 0.01). (D) Scatter plot of the SUSM vs. turns per minutes (estimated from IMU recordings, see Section 2.4 in S1 Text) for each animal. Along the $x$ axis (turns per minute), negative (resp. positive) values indicate ipsiversive (resp. contraversive) rotations relative to the lesion side. Each PD mouse ($n$ = 12) is shown twice: during the Parkinsonian phase (green dots) and during the LID phase (10–60 min period following L-DOPA injection, orange dots). The linear regression was computed for the LID phase. (E) Comparison of the SUSM before L-DOPA treatment (weeks 2 to 4) and the turns per minute during the LID phase (10–60 minutes following L-DOPA injection) for PD mice ($n$ = 12), showing that the SUSM during the Parkinsonian phase is predictive of the amount of contraversive rotations induced by L-DOPA. The data underlying panels C, D, and E can be found in S2 Data.

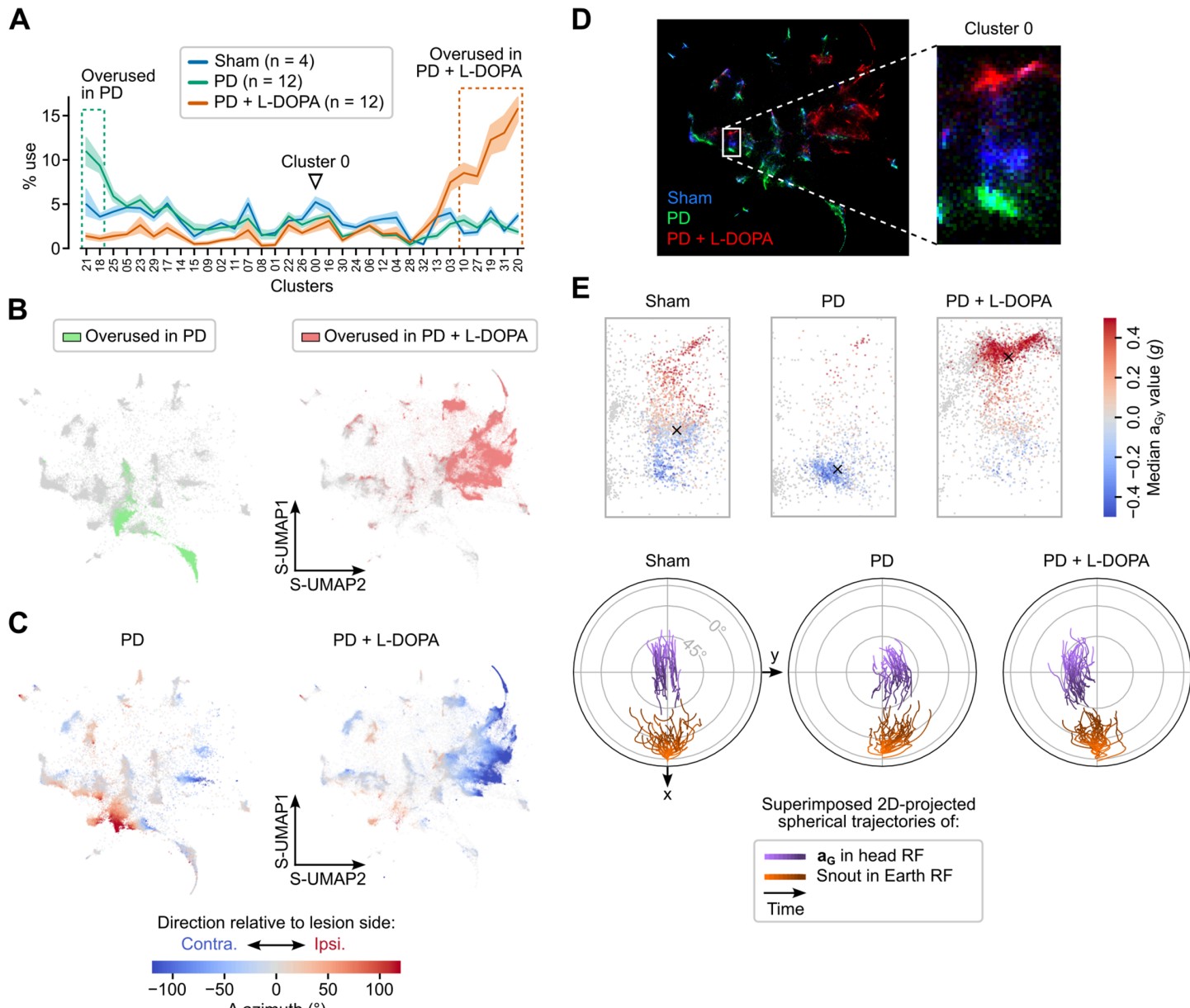

**Fig 8. DISSeCT captures both coarse and fine-grained changes in a mouse model of Parkinson's disease and levodopa-induced dyskinesia.** (A) Per-mouse average percentage use of individual clusters in sham (blue curve) vs. PD animals before (green curve) and after (orange curve) L-DOPA administration. Shaded areas represent 95% confidence intervals. Dashed rectangles indicate clusters exhibiting an increased use of more than 5% compared to sham mice (overused clusters) for PD mice before (green rectangle) and after (orange rectangle) L-DOPA injection. (B) S-UMAP embeddings highlighting segments belonging to overused clusters (defined in panel A) in PD mice before (left, green) and after (right, orange) L-DOPA injection. Gray points represent segments belonging to other clusters. (C) S-UMAP embeddings showing segments color coded according to the change of head azimuth. Positive (resp. negative) values represent ipsiversive (resp. contraversive) rotations relative to the lesion side. (D) Right: Superimposed S-UMAP embeddings for sham (blue), PD (green) and PD under L-DOPA (red) mice. Left: close-up view of cluster 0 (indicated by an arrowhead in panel A), which corresponds to upward head movements. (E) Left: close-up view of cluster 0 in the S-UMAP embedding space, with segments color coded according to the median value of $a_G$ along the interaural axis. Negative (resp. positive) values indicate a roll toward (resp. away from) the lesion side. Right: corresponding joint visualization of the trajectories of $a_G$ in the head reference frame (purple) and of the animal's heading in the Earth reference frame (orange) for the top 30 segments of cluster 0.

L-DOPA (Fig 8E). This example illustrates how DISSeCT can be used to detect and single out subtle changes in motor patterns.

## Discussion

IMUs are becoming an increasingly popular tool for motion tracking in neurophysiological research, yet their potential for parsing rodent behavior remains largely untapped. Unsupervised approaches, which do not rely on human labels to uncover data structure, are essential to fully assess this potential. Drawing inspiration from previous work on accelerometer or GPS data collected from free-roaming animals [85–87] and egocentric videos recorded in a clinical context [88], we developed DISSeCT, a two-step unsupervised segmentation and clustering pipeline specifically designed for rodent IMU data.

Similar to current video-based unsupervised behavioral mapping procedures [13,15,16], DISSeCT was able to distinguish the main categories of rodent activities (Fig 1), while also delivering fine-grained information on their various components. Our pipeline mapped the full repertoire of rapid 3D head orienting movements (Fig 2), offering a convenient alternative to methods based on threshold crossing detection in gyroscope measurements [39]. Interestingly, the duration and angular speed of the corresponding motifs co-varied to produce comparable head angular displacements (Fig 2B), suggesting a default angular set point for orienting movements. DISSeCT also successfully isolated episodes of locomotion, characterized by a rhythmic $\approx 6\,\mathrm{Hz}$ pattern in head-vertical acceleration (Fig 3). A similar signature at 8–10 Hz, scaling with animal speed (50–100 cm/s) was previously described in running rats [28]. The lower frequency observed here is probably related to the lower speed (10–15 cm/s) imposed by the restricted size of our arena. Inspection of a slow locomotion cluster revealed a $\approx 10\,\mathrm{Hz}$ sniffing signature distributed across specific axes (Fig 3D), and whose angular speed component has already been reported [31]. This signature, found in several other clusters (Fig 4), could be used in the future as a standalone metric to assess the level of olfactory exploration. Finally, grooming activities emerged as a family of clusters with diverse time-frequency properties, uniquely characterized by an uncoupling of the correlation between roll and yaw rotations from the head pitch angle (Fig 5A). These results demonstrate that DISSeCT is indeed a relevant approach for in-depth IMU data mining applied to rodent behavioral mapping.

A central question in behavioral mapping pipelines is whether a learned model can generalize to new observations. In our case, the learned GMM parameters—centers, covariance matrices, and mixture weights—are inherently influenced by IMU placement, since differences in sensor positioning and orientation can alter the comparability of extracted features. Consistent sensor placement is therefore critical for reliable embedding and clustering across animals or experiments. Similar constraints apply to pose-based models, which typically generalize only across animals of comparable size and anatomical proportions.

Generalizability can also be limited by variation in environment, task structure, or animal state, each of which may reshape the behavioral repertoire. Under stable conditions, it is generally reasonable to embed datasets from comparable animals within a shared model, as we did with our rat dataset. Likewise, a model trained on one dataset may be applied to another. However, this strategy becomes less reliable as experimental conditions diverge. For instance, in rodents, larger arenas promote the spontaneous expression of fast gaits such as galloping and bounding [89]. In such situations, novel behavioral motifs may be incorrectly mapped to clusters or syllables of a previously trained model, making model retraining the safer option.

Our framework provides tools to address this problem. Provided that the CPD penalty parameter is held constant, dataset similarity can be first assessed qualitatively through a

common segment embedding computed from the feature space (S14 Fig). The suitability of a trained GMM for a new dataset can then be quantitatively evaluated using cross-likelihood analysis (Fig 7C). These analyses do not prescribe a single outcome but instead support the experimenter in judging whether the existing model is sufficient or whether retraining is warranted. In the latter case, we recommend using the BIC to optimize the number of mixture components and select the most appropriate covariance structure (S4 Fig).

Future attempts to enhance model generalizability could focus on training a "global" GMM on a diverse, multi-condition, and well-characterized reference dataset. A key challenge of this strategy is that rare behaviors may not be assigned a dedicated Gaussian mixture component. The reference dataset should therefore be carefully constructed and weighted to adequately represent the full behavioral repertoire, including infrequent motifs. The resulting model should then be systematically evaluated on datasets collected across laboratories to test its robustness and transferability.

When rerunning the entire pipeline becomes necessary, the associated computational cost and effort of re-evaluating model output are important considerations. Several factors mitigate these drawbacks. First, the exploitation of new DISSeCT runs will benefit from our detailed characterization, which reliably anchors core behaviors (e.g., locomotion, sniffing, grooming, extended rearing) to specific kinematic signatures and regions of the feature space. Second, we advise maintaining a minimal parallel video stream for post hoc validation of cluster content; our companion tool, INSpECT, streamlines this process by enabling rapid browsing and labeling of large numbers of video snippets. Third, DISSeCT's efficiency and modest computational requirements, discussed below (see also S4 Table), significantly reduce the burden of multiple reruns. They also enable iterative exploratory analyses such as adjusting the CPD penalty to probe behavioral structure at different temporal scales within selected recording periods.

Cluster inspection revealed that ethologically distinct motifs may sometimes share similar head kinematic properties, such as upward head movements and rearing initiation. Using a post-hoc categorical HMM to model the symbolic sequence resulting from segment clustering (Fig 6), we show that some of these ambiguities can be resolved if the corresponding motifs occur within distinct identifiable sequences (S11 Fig). Other solutions to improve the separation of similar motifs include fitting GMMs to the corresponding subsets of segments using tailored features or employing hierarchical clustering methods. Missing body kinematic information could be obtained using additional IMUs, either subcutaneously implanted [47,51] or embedded into jackets worn by animals, keeping in mind that sensor position and orientation may vary over time [90]. Since maintaining a minimal concurrent video stream is a practical option for rapid inspection of clusters, as discussed above, basic video-derived features could be integrated into DISSeCT to help resolve kinematic ambiguities. The potential utility of this approach is supported by our observation that cluster-wise purity scores were more robust to random sampling for KpMS syllables—derived from video—than for DISSeCT clusters (S12C to S12G Fig). This hybrid strategy could enhance the pipeline's discriminative power without substantially increasing experimental complexity. It would also allow social context—such as inter-animal proximity—to be incorporated into IMU-based analyses involving multiple interacting animals. In this setting, wireless IMU recordings would provide fully individualized and independent per-animal measurements [30], each of which could be processed separately using our pipeline. Provided that individual identities can be reliably tracked from video, this combined optical-IMU framework could enable the discovery of social interaction clusters, marked for example by high sniffing activity and close proximity to a conspecific individual.

In rodent research, IMUs have primarily been used for automatic behavioral scoring, typically relying on fixed-length data segmentation combined with supervised classifiers [26,51, 52]. To our knowledge, only two main approaches have employed IMU recordings for unsupervised rodent behavioral mapping, with the goal of linking the resulting cluster sequence to concurrent recordings of neuronal activity. In one case, feature extraction over fixed-length windows was performed on combined IMU and video data, and behavioral motifs were uncovered by applying affinity propagation to a pairwise window similarity matrix [33,91]. In the other, IMU data were used primarily to refine 3D head orientation estimates within a broader pose-based behavioral mapping framework [16]. In contrast, our approach represents, to our knowledge, the first unsupervised rodent behavioral mapping pipeline that relies exclusively on inertial recordings.

Our pipeline also differs in several ways from existing video-based, unsupervised rodent behavioral analysis frameworks, which can be broadly grouped into two categories. The first category relies on extracting features from fixed-length video segments and embedding them into 2D maps, where recurring behavioral patterns manifest as high-density areas. This strategy is employed in pipelines such as MotionMapper [70], B-SOiD [77], and DANNCE [13]. While effective for broad behavioral mapping [15,16,18], this approach is not designed to precisely delineate behavioral motifs in time. Additionally, its drastic dimensionality reduction prior to clustering potentially diminishes its discriminative power [71]. In contrast, DISSeCT uses CPD to accurately locate behavioral transitions and performs clustering in a high-dimensional space, minimizing information loss. The second category aims at explicitly modeling temporal behavioral dynamics using graphical models. For example, Keypoint-MoSeq [17] combines autoregressive HMMs with switching linear dynamical systems to jointly infer behavioral motif limits and identities from PCA-reduced keypoint trajectories. Another pipeline, VAME [63], occupies a middle ground between these two families: it first uses representation learning to construct a lower-dimensional embedding of pose trajectories from fixed-length windows, then applies an HMM to simultaneously segment and cluster the resulting latent time series.

Here, we propose an alternative approach in which time series segmentation is decoupled from the clustering step, with two main objectives. The first is to improve computational efficiency. In our conditions, a full CPU-based DISSeCT analysis ran approximately five times faster than a GPU-based KpMS analysis (see S4 and S5 Tables), even before accounting for the additional time and resources required for video-based pose estimation. Both methods operated on a similar number of dimensions (DISSeCT used a 9-dimensional inertial time series, while KpMS modeled an 8-dimensional latent time series), though a key difference was that IMU data were sampled 10 times faster (300 Hz vs. 30 Hz). Given that KpMS's computing time and GPU memory usage scale linearly with dataset size, applying it directly to IMU data would increase runtime and memory requirements by a factor of at least 50. This makes such an approach impractical for routine use in large-scale or long-duration behavioral recordings, unless significant computational resources—such as high-performance GPU clusters—are readily available. The second objective, discussed below, is to maintain a modular and adaptable pipeline.

DISSeCT's modular design allows for the integration of alternative clustering methods or iterative processes, such as additional rounds of feature extraction and segment clustering for finer-grained mapping of specific data subspaces. Automated scoring of user-defined activity patterns could also be achieved using a supervised learning algorithm downstream of the CPD step. Regardless of whether segments are analyzed using supervised or unsupervised methods, feature design and selection remains a critical component of the pipeline. A promising direction involves formalizing feature extraction and dimensionality reduction through

representation learning, an approach that seeks to uncover the most informative nonlinear embeddings prior to the modeling stage. For instance, automatic feature engineering was achieved using variational autoencoders in VAME, a recent pose-based, unsupervised analysis pipeline [63]. This technique, however, is currently constrained by its reliance on fixed-length input windows. Extending such methods to accommodate variable-length segments, such as those derived from CPD, would require further methodological advances, potentially leveraging transformer-based architectures [92].

The combined resolution and versatility of IMU recordings open up a wide range of potential applications for our pipeline. By accurately detecting abrupt behavioral transitions and parsing brief behavioral motifs, DISSeCT provides the temporal resolution required for neuroethological studies aimed at uncovering the neuronal mechanisms underlying natural behavior [2]. One particular area where our pipeline could significantly advance the state-of-the-art is in analyzing behaviors that are challenging to accurately assess via video. Self-grooming, for example, is a complex behavior associated with contorted postures where IMUs offer significant advantages over video recordings for detailed analysis [16,26,51]. DISSeCT successfully parsed the main components of self-grooming, shedding light on their distinct head kinematic signatures. While further refinement is needed to enhance the separation of grooming subcomponents, our approach introduces a powerful new method for automating the analysis of this elaborate syntactic behavior [83], with potential applications in the study of rodent models of neurological and psychiatric disorders [93]. Similarly, wet dog shake events were easily isolated due to their prominent and unique spectral signature (S5 Fig), accounting for the vast majority of outlier segments (Fig 1C). Their study is relevant in the context of rodent models of diseases including acute seizures [94], morphine abstinence [95], and nicotine withdrawal [96].

DISSeCT's use of a GMM layer facilitates direct comparison between experimental groups. The probabilistic formulation of GMMs enables cross-likelihood evaluation, where a model trained on one dataset can be used to assess how well it explains a separate set of observations, and vice versa. We illustrated this approach in a mouse model of Parkinson's disease and levodopa-induced dyskinesia, where our pipeline enabled the identification of both global and subtle motor alterations (Figs 7 and 8). Alternatively, GMMs fitted independently on different datasets can be compared directly using entropy-based measures such as the Jensen–Shannon divergence, which quantify differences in the structure of behavioral distributions. By transforming high-resolution kinematic data into interpretable and comparable probabilistic representations, our pipeline offers a promising new solution for deep behavioral phenotyping. Potential applications could be found in the detection, quantification, and characterization of behavioral changes—both within and between individuals—across natural processes (e.g., development or aging) or in response to experimental interventions (e.g., in disease models, learning protocols or environment manipulations).

As outlined in the Introduction, inertial recordings bypass many of the common limitations of video recordings, but come with their own constraints—chiefly, the requirement for surgical implantation. Deploying IMUs in large cohorts can therefore entail substantial time investment compared to video-based strategies. This limitation is less relevant in neurophysiological studies, where animals are routinely implanted with recording devices that often already include an IMU. A key advantage of IMUs is their ability to track animals in large, naturalistic environments, including occluded or hidden spaces. Transitioning to such settings requires wireless IMUs, which introduces a necessary trade-off between sampling rate (a key determinant of power consumption) and battery life. Although low-power, data-logging IMUs offer a promising solution, they have yet to be adapted for use in small animals such as mice. Another limitation is that, unlike video, inertial signals are not readily interpretable

by an untrained observer. To address this, both our previous study [42] and the present work introduce a set of metrics and graphical tools to aid in the exploration and interpretation of IMU data. Based on the output of our pipeline, we also highlight distinct head kinematic signatures that reliably map onto specific behaviors. Together, these resources aim to support and standardize future IMU-based behavioral investigations.

In conclusion, we propose and validate a novel unsupervised framework for the analysis of rodent inertial data that combines kernel change-point detection and probabilistic modeling. By leveraging the unique properties of IMUs, our pipeline enables access to an underexplored region of the "behavioral science space" [97], where high-resolution, fine-grained behavioral analysis can be conducted with minimal environmental constraints and computational cost.

## Material and methods

### Ethical approval

This project was reviewed by the "Charles Darwin" Ethical Review Board (CEEA-005) and approved by the department responsible for overseeing the use of animals for scientific purposes within the French Ministry of Higher Education, Research, and Innovation (AFiS unit). Approval was granted under project license number APAFIS #29793-202102121752192 v3.

### Animals

Four adult male Long-Evans rats (Janvier Labs) and sixteen RJOrl:SWISS mice (CD-1, eight males and eight females, Janvier Labs) were used in this study. They were group-housed under standard conditions (12-hour non-inverted light/dark cycle, 21–24 °C, 50–60% humidity) and provided with unrestricted access to food and water along with appropriate cage enrichment such as cardboard tunnels, shredded paper, and dental cotton rolls.

### Surgical procedures

**Wired IMU fixation in rats.** A wired IMU measuring $5.0 \times 7.5 \times 3.0$ mm and weighing 0.3 g (mptek.fr), housing a 9-axis digital inertial motion sensor (MPU-9250, Invensense), and operated via a dedicated USB acquisition board (mptek.fr) was used for rat recording sessions. The IMU featured a $2 \times 4$ pin header (TSM-104-01-S-DV, Samtec Inc.) used to connect the device to its counterpart female socket header (NPPC042KFMS-RC, Sullins Connector Solutions), which was permanently affixed to the animal's skull through a surgical procedure outlined below. Animals underwent surgery at six weeks of age and recovered for one week before recording.

The connector implantation surgery was carried out as follows. Preemptive analgesia was ensured by subcutaneously injecting buprenorphine (0.01 mg/kg) thirty minutes before the procedure. Anesthesia was induced with 3% isoflurane in oxygen delivered at 3 L/min. The animal was then positioned in a stereotaxic apparatus, with its eyes protected by ophthalmic gel. Throughout the remainder of the procedure, body temperature was kept at 37 °C using a homeothermic monitoring system, while anesthesia was maintained with 1% isoflurane delivered at 0.5 L/min. The animal's scalp was shaved and cleansed with povidone-iodine followed by 70% ethanol. Prior to incising the scalp, a subcutaneous injection of lidocaine (2%) was administered. The periosteum was carefully scraped with a scalpel blade, followed by application of a 3% hydrogen peroxide solution. The female socket header, held upright using one arm of the apparatus, was then affixed to the cranial surface using self-curing dental adhesive (Super-Bond C&B, Sun Medical). Finally, the skin around the connector was sutured both anteriorly and posteriorly, and the animal was placed on a heating blanket while its recovery was monitored. Once fully alert and mobile, the rat was returned to its home cage.

**6-OHDA lesion and wireless IMU fixation in mice.** A wireless IMU (WEAR wireless sensor v2.1, Champalimaud Scientific Hardware Platform) measuring 14 × 6 × 26 mm and weighing 1.85 g, housing a 9-axis digital inertial motion sensor (MPU-9250, Invensense), and operated via a dedicated USB acquisition board (WEAR basestation, Champalimaud Scientific Hardware Platform), was used for mouse recording sessions. The IMU was mounted on the head using a 1 × 4 pin header (M52-040023V0445, Harwin Inc.) on the IMU side and its counterpart female socket header (M52-5000445, Harwin Inc.) on the animal side (see below).

Unilateral nigrostriatal dopaminergic denervation was achieved at 9–10 weeks of age by injecting 6-hydroxydopamine hydrochloride (6-OHDA, Sigma-Aldrich) into the medial forebrain bundle (MFB) as previously described [98,99]. A subcutaneous dose of buprenorphine (0.1 mg/kg) and an intraperitoneal dose of desipramine (25 mg/kg, Sigma-Aldrich) were administered thirty minutes before initiating the procedure. The initial steps of the surgery, from anesthesia induction to skull preparation, were similar to the rat connector implantation protocol described above. For the Parkinsonian model (PD mice), the left MFB (−1.2 mm AP, 1.25 mm ML and −4.7 mm DV relative to Bregma) was injected with 1 μL of a 6-OHDA solution (3.2 μg/μL free-base concentration in 0.02% ascorbic acid, Sigma-Aldrich); whereas for control animals (sham mice), an equivalent volume of the vehicle solution (0.02% ascorbic acid) was administered. The injection was performed with a Hamilton syringe at a rate of 0.3 μL/min, with the syringe maintained in situ for two minutes before and four minutes after injection. The female socket header used for IMU fixation, held upright using one arm of the apparatus, was then affixed to the cranial surface using self-curing dental adhesive (Super-Bond C&B, Sun Medical). The skin around the connector was sutured both anteriorly and posteriorly, and the animal received a subcutaneous injection of Meloxicam (5 mg/kg, Metacam, Boehringer Ingelheim) and of a glucose-saline solution (Glucose 5%, 10 mL/kg, Osalia). Throughout the recovery phase, the mouse was placed on a heating blanket while being closely monitored. Once fully alert and mobile, it was returned to its home cage. To mitigate the risk of post-surgical complications and distress, mice received daily injections of glucose-saline solution and were nourished with a blend of Blédine (Blédina, Danone) and concentrated milk for a period spanning 6 to 10 days post-surgery, resulting in a 100% survival rate.

## Data acquisition

**Synchronized inertial and video data acquisition for quantifying rat behavior.** Rat recordings started one week after connector implantation. Once equipped with the wired IMU, rats were placed in a 70 × 70 arena with transparent glass floor and transparent PMMA walls, allowing for video monitoring from multiple angles, including from underneath, similarly to previous studies [47,77,100]. Inertial signals were acquired at 300 Hz with accelerometer and gyroscope ranges set at ± 2 g and ± 1000°/s, respectively. Video recordings were captured using five off-the-shelf industrial cameras (DMK 37BUX273, The Imaging Source) connected via USB to an acquisition computer. Frame acquisition was triggered at 30 Hz by an Arduino Uno running a custom pulse-width modulation script. These triggers were simultaneously recorded by the IMU acquisition board to precisely timestamp video frame acquisition within the IMU data stream. In our conditions, occasional frame drops were observed, at a frequency which gradually increased during continuous acquisition. We found that delivering triggers in one-minute chunks (comprising 1,800 frames), separated by three-second pauses, and saving the corresponding videos as separate files, drastically reduced the occurrence of frame drops to 0.023% (0.39 ± 0.65 missing frame per chunk).

**Synchronized inertial and video acquisition for quantifying behavior in a mouse model of Parkinson's disease and levodopa-induced dyskinesia.** Considering that 6-OHDA injection and connector implantation were conducted at day 0, IMU recordings started on day 10. Once equipped with the wireless IMU, mice were placed in a $30 \times 35$ arena with transparent PMMA floor and walls. Video recordings were captured using two off-the-shelf industrial cameras (DMK 37BUX273, The Imaging Source) positioned below and on the side of the arena. Synchronized acquisition of video and IMU data was carried out using dedicated acquisition hardware (WEAR basestation, Champalimaud Scientific Hardware Platform) controlled via a custom pipeline programmed in Bonsai [101]. Videos were acquired at a rate of 50 Hz, while IMU data were recorded at 200 Hz with accelerometer and gyroscope ranges set at $\pm 2\,\text{g}$ and $\pm 1000°/\text{s}$, respectively.

Recordings were conducted over a five-week period, with one recording session per week (Fig 7A). The first three sessions (days 10, 17, and 24) lasted 30 minutes each. Starting on the 24th day, mice received daily intraperitoneal injections of L-3,4-dihydroxyphenylalanine methyl ester hydrochloride (L-DOPA or levodopa, 3 days at 3 mg/kg and then 6 mg/kg; D1507, Sigma-Aldrich) and benserazide hydrochloride (12 mg/kg; B7283, Sigma-Aldrich), a peripheral decarboxylase inhibitor. The final two recording sessions were conducted on days 31 and 38, starting 10 minutes before and extending 60 minutes after levodopa injection.

## Markerless 3D pose estimation

In the case of rat video recordings, the use of five different viewpoints enabled 3D pose estimation. Using the DeepLabCut toolbox [6], a unique deep neural network was trained to recognize twelve anatomical keypoints from all viewpoints, providing an independent 2D pose estimate for each video. The training dataset consisted of 1 950 hand-labeled frames selected to encompass all different viewpoints and a wide range of animal positions and postures within the arena. Additional details regarding the parameters employed can be found in S1 Table. Before each session, the intrinsic and extrinsic parameters of the cameras were estimated by recording a video of a ChArUco board moved throughout the arena, and applying the calibration pipeline available in Anipose [12]. This pipeline combines checkerboard detection using OpenCV with an optimization procedure aimed at minimizing reprojection errors.

Spatio-temporal filtering is a crucial step in obtaining high quality estimates of 3D pose dynamics [12,47,102]. For this purpose, we used the filtering pipelines integrated into Anipose. Median filtering was first applied to the five independent 2D pose estimates to alleviate the impact of sudden keypoint jumps. Triangulation was then performed using camera calibration and spatio-temporal regularization, taking advantage of the continuity and relative smoothness of animal movements, as well as the stability of body part lengths over time. Finally, a 3D median filter was applied to the resulting 3D pose estimate to reduce keypoint jitter. A summary of the parameters used for triangulation and filtering with Anipose is provided in S2 Table.

## Pre-processing of IMU data

**Subtraction of sensor offsets.** The estimation of sensor offsets followed a procedure described previously [42]. For each IMU, data was collected while placing the device in a set of 50 different static positions using a custom robotic arm. Keeping only the periods of immobility, accelerometer offsets were estimated by minimizing the average squared difference between the Euclidean norm of offset-corrected acceleration measurements and 1 g. Gyroscope offsets were computed as the median gyroscope values observed during these periods of

immobility. Accelerometer and gyroscope offsets were then subtracted from acceleration and angular velocity measurements, respectively.

**Estimation of head tilt.** Head orientation relative to gravity (head tilt) can be accurately estimated from rodent IMU data using the appropriate Attitude and Heading Reference System (AHRS) filtering algorithm. For each IMU data point, an extended Kalman filter [103] with a previously validated set of parameters ($var_{acc}$ = 0.002, $var_{gyr}$ = 0.75) [42] was used to compute a unit quaternion describing sensor orientation relative to an arbitrary, gravity-polarized Earth reference frame (absolute reference frame). This quaternion represents a 3D rotation which, when applied to this absolute reference frame, aligns it with the sensor's orientation. To estimate head orientation relative to gravity (head tilt, i.e., the orientation of the gravitational acceleration vector in the head reference frame), the inverse rotation, corresponding to the conjugate quaternion, was applied to the unit vector (0   0   1).

## Change-point detection

**Algorithm.** Pre-processed IMU data were segmented by applying the PELT (Pruned Exact Linear Time) change-point detection method to z-scored gravitational acceleration and angular velocity time series (non-gravitational acceleration was not used here because of its lower signal-to-noise ratio). This algorithm leverages dynamic programming to find the optimal number and location of change points in time series by minimizing a cost function (see Section 1 in S1 Text), while maintaining a linear computation time [65]. Specifically, we used a kernel-based change-point detection algorithm [64] relying on a Gaussian kernel, which offers the advantage of being characteristic; meaning that changes in the mean of the embedded data correspond to changes in the data distribution [69,104,105] (see Section 1 in S1 Text).

**Selection of the penalty parameter.** Penalty for segmenting IMU data was set to $\lambda$ = 14, yielding segments with a median duration of 377 ms, to align with the typical duration of individual elements identified by MoSeq, a depth video-based unsupervised method used to parse mouse behavior [62]. This duration also matched the timescale of the finest behavioral motifs identifiable by a human observer (see S2 Fig).

## Segment clustering

**Feature extraction.** A total of 240 features were extracted for each segment, effectively capturing postural and kinematic aspects of the animal's activity within the segment. These features included statistics on the signals, parameters derived from head tilt and changes in head azimuth, as well as coefficients from continuous wavelet transforms of angular velocity and non-gravitational acceleration data (see Section 2 in S1 Text).

**Outlier removal.** A small subset of segments exhibited outlier feature values, leading to convergence issues during clustering. To address this, a threshold of 35 was applied to absolute robustly rescaled feature values, obtained through median subtraction and division by the interquartile range. Segments exceeding this threshold were excluded from the analysis. These outliers consisted of extended periods of immobility (17%) and high magnitude events corresponding either to vigorous movements (21%, typically > 1 g of non-gravitational acceleration and > 500°/s of angular speed) or to wet dog shakes [106] (with the norm of non-gravitational acceleration and angular velocity exceeding 2 g and 1000 °/s, respectively). The latter type of outlier is characterized by a high amplitude 16–20 Hz oscillatory regime whose wavelet transform signature artificially contaminates neighboring segments due to edge effects (S5 Fig). Consequently, segments immediately before and after wet dog shake

events were also excluded from the analysis. This curation procedure resulted in the exclusion of 1.7% of all segments, representing 3.7% of the total recording duration.

**Gaussian mixture model clustering.** After having excluded outlier segments as described above, features were z-scored and projected in a lower-dimensional space via PCA. The number of principal components was determined through cross-validation [107] (S4A Fig). A clustering step was then executed on the resulting projection using `scikit-learn`'s GMM implementation. Model selection and hyperparameter tuning were conducted through grid search with the aim of maximizing the Bayesian information criterion (BIC) [108]. The grid search encompassed covariance type (full, diagonal, tied, spherical) and number of clusters (ranging from 5 to 100), employing 10 random initialization for each hyperparameter set (S4B Fig), yielding 48 mixture components with full covariance matrices. In our terminology, *clusters* are groups of segments assigned to the same most likely generating mixture component, as determined by posterior probabilities. Accordingly, this hard assignment step yielded 48 clusters, each corresponding to a mixture component in the GMM. These clusters are defined in the PC space and do not correspond *a priori* to behavioral categories. Their ethological relevance is assessed in subsequent analyses. More information on the procedure is available in section 4 of S1 Text.

**Ablation of immobility periods prior to modeling.** To evaluate whether a CPD-based segmentation strategy provided a better fit than a simple fixed-length windowing approach, we performed an ablation of immobility periods prior to Gaussian mixture modeling. Fixed-length windows falling within immobility periods were identified using a threshold of 12 °/s on the standard deviation of angular speed (i.e., the norm of gyroscope measurements), as illustrated in S2D Fig. This procedure excluded 7.3% of all windows (3 222 out of 44 426), totaling 2255.4 s. Because immobility segments isolated by CPD often corresponded to long periods of immobility briefly interrupted by small movements, the same threshold-based exclusion could not be directly applied. Instead, we ranked CPD segments by angular speed standard deviation and incrementally removed the lowest-activity segments until their total duration matched that of the excluded fixed-length windows. This resulted in the removal of 2.05% of all CPD segments (908 out of 44 322), corresponding to a total duration of 2255.1 s. After feature extraction and PCA, we fit a series of 10 independent GMMs over a range of component numbers ($K$), using either the full or ablated sets of segments/windows. For each $K$, the average log-likelihood was computed on the corresponding set of segments/windows.

**2D embedding of clustered data.** For visualization, z-scored features associated with individual segments were projected onto a 2D space using supervised UMAP embedding [109] (with parameters: $n\_neighbors = 8$, $target\_weight = 1e - 9$, $min\_dist = 1e - 3$). This method incorporates cluster label information to guide the embedding process, ensuring that points belonging to the same cluster are placed closer together, while still preserving the overall structure of the data. Feature values computed on outlier segments, excluded from the clustering step, were also included in this embedding step, albeit ignoring the associated label.

**Sparse clusters.** The dispersion of individual mixture components was evaluated by calculating the trace of their covariance matrix. Clusters associated with the three mixtures exhibiting the highest trace values were considered as sparse clusters (S7A Fig). Collectively, these clusters accounted for 2.1% of the total number of segments and for 1.0% of the total recording duration. Examination of these clusters revealed that they comprised heterogeneous segments, corresponding to occasional brief and vigorous movements (approaching 2 g of non-gravitational acceleration and 1000 °/s of angular velocity), and which appeared highly dispersed on the UMAP embedding (S7B Fig). The presence of such clusters is expected, as GMMs tend to assign rare and/or outlier observations to a few highly dispersed and weakly informative clusters [110].

### *Ad hoc* metrics

**Per-segment average head roll and pitch angles.** Within the head's reference frame, gravitational acceleration can be conceptualized as a unit position vector, more specifically as a point on a head-centered sphere of radius 1 g that fully represents head tilt. Due to the non-linearity of trigonometric functions, calculating the average head tilt over a certain period of time is not as straightforward as averaging the coordinates of this point. Instead, the average head tilt can determined as the centroid (estimated mean direction) of a von Mises-Fisher distribution fitted to the various positions assumed by this point on the unit sphere. Noting this centroid $\boldsymbol{\mu} = \begin{bmatrix} \mu_x & \mu_y & \mu_z \end{bmatrix}$, the corresponding average head roll and pitch angles are computed as the opposite values of the polar coordinates of $\boldsymbol{\mu}$:

- $\mu_{roll} = -arctan2(\mu_y, \mu_z)$
- $\mu_{pitch} = -arctan2(-\mu_x, \sqrt{\mu_y^2 + \mu_z^2})$

**Detection of immobility.** Angular speed was computed as the Euclidean norm of offset-corrected gyroscope measurements and then smoothed using a Gaussian kernel ($\sigma = 100$ ms). The animal was considered immobile when this value was below 20 °/s.

**Sniffing score.** To identify segments displaying the $\approx 10$ Hz sniffing signature primarily observed in the naso-occipital non-gravitational acceleration (noted $a_{nG,x}$) and pitch angular velocity (noted $\omega_y$), we designed an *ad hoc* sniffing score (noted $\Phi$). This metric was constructed empirically using the maximal wavelet power (noted $W$) across all available frequencies (1–20 Hz) or frequencies largely above (> 14 Hz), below (< 6 Hz) or within the band of interest (8.5–11 Hz):

$$\Phi = \alpha \cdot \frac{W_{8.5-11\,Hz}^{a_{nG,x}} + W_{8.5-11\,Hz}^{\omega_y}}{W_{fullrange}^{\omega_x} + W_{<6\,Hz}^{a_{nG,x}} + W_{>14\,Hz}^{a_{nG,x}} + W_{fullrange}^{a_{nG,y}} + W_{fullrange}^{a_{nG,z}} + 100}$$

where $\alpha$ is a factor that penalizes $\Phi$ for segments associated with rapid head reorientation or large roll angles, which often exhibit non-sniffing related 8–10 Hz signatures. Brief reorientations can indeed last approximately 100 ms, thereby displaying a $\approx 10$ Hz time-frequency signature, while substantial roll angles are linked to body grooming, an activity with a typical $\approx 4$ Hz pattern that may manifest a notable harmonic at about 8 Hz. This factor was computed as follows:

$$\alpha = \frac{1}{1 + \eta} \cdot \frac{1}{1 + |\mu_{roll}|}$$

with $\eta$ representing the heading change rate in °/s (the angle between heading estimates at the beginning and end of a segment, divided by its duration) and $\mu_{roll}$ the roll of the average tilt in radians.

## 2D visualization of per-segment head attitude and heading trajectories

To visualize 3D head orientation during individual segments, we overlaid the trajectories of head-tilt (pitch and roll) and 3D-heading (pitch and azimuth). Head-tilt was represented using the coordinates of the Earth's vertical in the head reference frame, estimated by the gravitational component of acceleration signals obtained through AHRS filtering of IMU data (see Methods). The estimate of 3D-heading was derived by applying the rotation from the head reference frame to a gravity-polarized Earth reference frame, achieved via AHRS

filtering, to the x-axis of the sensor's reference frame. The resulting vector was then rotated along the vertical axis to set the first sample of the segment to an azimuth of 0 in the Earth reference frame (see Section 2.2 in S1 Text).

These 3D time series of unit-norm vectors were projected to two dimensions using a Lambert Azimuthal-Equal Area projection, centered either on the *North Pole* (i.e., (0,0,1)) or *South Pole* (i.e., (0,0,−1)) of the respective reference frames. The trajectories were then superimposed on a single plot using the cartography module `cartopy`.

### Post hoc categorical HMM

A categorical hidden Markov model was fitted to the sequence of clusters obtained from GMM modeling, using the implementation provided by `hmmlearn`. Segments deemed outliers, as defined earlier, were treated as an independent cluster. Hyperparameter optimization was conducted via BIC maximization, employing ten independent fits per hyperparameter value, to determine the optimal number of states. More information on the procedure is available in Section 5 of S1 Text.

### Keypoint-MoSeq analysis

To analyze our 3D rat pose dataset using the Keypoint-MoSeq (KpMS) pipeline [17], we first optimized the stickiness hyperparameter ($\lambda$) to achieve a median syllable duration of 367 ms, aligning with previously reported values [17,62] and with the median segment duration derived from change-point detection applied to IMU data (377 ms; see above). We then fit six independent KpMS models to the dataset and selected the best-performing one based on the expected marginal likelihood (EML) score [17]. All parameters used for running KpMS are listed in S3 Table. Computation times for the GPU and CPU versions of KpMS are reported in S5 Table.

Behavioral syllables obtained from KpMS were compared to DISSeCT clusters by aligning each video frame to the corresponding IMU sample. This alignment was based on detecting the rising edge of the trigger pulse that initiated video frame acquisition, as recorded in the IMU data stream. A contingency matrix (S13 Fig) was then constructed by counting the number of co-occurring IMU-video frame pairs assigned to each KpMS syllable and DISSeCT cluster.

### Inspection and labeling of video segments

**Software.** To expedite the inspection and labeling of video snippets corresponding to selected DISSeCT segments or KpMS syllable instances, we contracted AquiNeuro to develop custom software. This software, named INSpECT (Interface for Navigating Spates of video Excerpts and Categorizing Them), features a graphical user interface and keyboard shortcuts for navigating a series of multi-camera video snippets and assigning user-defined labels to each (see Data availability).

**Inspection and labeling.** Given a set of video snippets corresponding to selected DISSeCT segments or KpMS syllable instances, an observer was tasked with assigning a unique primary category to each one (choosing from the following options: "orienting", "rearing", "grooming", "locomoting", "idle" or "other"), along with an additional unconstrained secondary label providing a more detailed description of the ongoing behavioral activity. The observer was blind to the identity of the DISSeCT cluster or KpMS syllable from which each snippet originated. The category "orienting" was used for rapid changes in head orientation unrelated to rearing or grooming episodes. "Rearing" referred to situations during which the

animal stood on its hind legs with both forelegs above the ground. "Grooming" encompassed all types of self-cleaning activities. The category "idle" applied to situations where the animal stood immobile, occasionally accompanied by small twitches or head sways.

To quantify how consistently each cluster or syllable was associated with a specific behavior, we computed a *purity score* [63], defined as the proportion of snippets assigned to the most frequent category:

$$\rho = \frac{1}{|S|} \max_{c \in C} \left| \{ s \in S \mid \text{label}(s) = c \} \right|$$

where $S$ is a set of video snippets $s$ associated with a given cluster or syllable, and $C$ is the set of behavioral categories $c$.

Prior to snippet selection, the following eligibility criteria were applied: (1) a minimum duration of 7 video frames (233 ms) to ensure visibility of behavior by the observer, (2) DISSeCT segments had to be classified as non-outliers (see Methods) and fully contained within video chunks (see Methods), (3) KpMS instances had to originate from syllables accounting for more than 5% of the total recording duration. Applying these criteria yielded 41 005 eligible DISSeCT segments (out of 44 322) and 30 230 eligible KpMS syllable instances (out of 48 854).

Snippet selection was then performed in one of two ways: (1) by selecting the top one hundred most representative segments or instances per cluster or syllable, or (2) by randomly sampling 10% of all eligible segments or instances. For DISSeCT, the top one hundred segments per cluster were defined as those with the highest posterior probabilities assigned to that cluster. For KpMS, we used the best-performing model to estimate frame-wise marginal probabilities over syllable labels. Consecutive frames with the same highest-probability label were grouped to form new syllable instances. The instance-level probability was computed as the average of the frame-wise probabilities and used to rank instances. For each syllable, the top one hundred instances with the highest scores were selected.

### Mapping of healthy and pathological mouse behavior

**Pipeline implementation.** Change-point detection was conducted on all available mouse IMU data. Similar to the approach taken with rat data, the penalty level was set to a value ($\lambda = 9$ here) resulting in a median segment duration of about 350 ms. The same set of features used for rats was extracted for each segment, and the identical procedure for detecting outlier segments was applied, leading to the exclusion of 0.7% of all segments from the clustering step. These segments primarily consisted of extended periods of immobility and accounted for 10.6% of the total recording duration.

Parameters for feature rescaling, PCA dimensionality reduction and Gaussian mixture modeling were solely fitted on sham (healthy) mouse recordings before levodopa treatment. Hyperparameter optimization based on BIC maximization yielded 33 clusters with full covariance matrices. Based on this model, cluster assignment was then carried out for all other non-outlier segments (from sham mice under levodopa treatment and PD mice in all conditions).

The final supervised UMAP embedding was fitted using only one out of three segments for computational efficiency and using the same hyperparameters as for rat data, and was then applied for all mice and conditions.

**Score under a sham model.** The GMM fitted on segments from sham mice prior to levodopa treatment served as the reference model for computing the log-likelihood of all

available segments. For any given segment, this value indicates how likely the segment is under the assumption that the mouse is healthy. The average log-likelihood across a set of segments, referred to hereafter as the *score under a sham model* (SUSM), was used as a measure of how well the corresponding IMU recording bout could be explained by the healthy behavioral model.

To prevent potential overfitting when evaluating sham animals, we adopted a leave-one-out approach: for each sham mouse, the SUSM was computed using a GMM trained on all other sham animals, excluding the test animal. This allowed for a more accurate estimate of model generalization within the sham group. In contrast, segments from PD and L-DOPA mice were evaluated using a single GMM trained on the full sham dataset, reflecting the use of this model as a normative reference. All GMMs used identical hyperparameters (number of components and covariance structure) as determined by BIC on the full sham data.

**Overlay of behavioral maps.** To overlay UMAP scatter plots (Fig 8D), we first transformed them into 2D histograms using a $500 \times 500$ grid. The bin values were then rescaled to fall between 0 and 1 by dividing each value by the 99.8[th] percentile of the histogram, with any values exceeding one clipped to one to mitigate the impact of outliers. The resulting histograms were visualized as an image using the red, green, and blue channels for PD-LDOPA, PD, and sham mice, respectively.

## Acknowledgments

We extend our gratitude to Ahmed Boughdiri and Stefano Bettani for their assistance in setting up the multi-camera video acquisition system, and to Idriss Tsayem Ngueguim for his help in rat implantation surgeries. We also thank Ana Margarida Pinto for her help in establishing the acquisition system used for mouse recordings, Adja Sissokho for her assistance with mouse experiments, and John Jacoby for his guidance in setting up Keypoint-MoSeq analysis. Additionally, we are grateful to Laurent Oudre, Éric Burguière, Ashesh Dhawale, and Bence Ölveczky for their valuable feedback on the project. Our thanks also go to the IBENS computing platform, fablab, and animal facility for their support.

## Supporting information

**S1 Text. Supporting methods.**
(PDF)

**S1 Table. Summary of the parameters used for obtaining 2D pose estimates with DeepLabCut.**
(PDF)

**S2 Table. Summary of the parameters used for obtaining 3D pose estimates using Anipose.**
(PDF)

**S3 Table. Summary of the parameters used for running Keypoint-MoSeq on our dataset.**
(PDF)

**S4 Table. Computation times for the different steps of DISSeCT**, measured on the same workstation used for running DeepLabCut, Anipose, and Keypoint-MoSeq (Dell Precision 5820 Tower with an NVIDIA Quadro RTX 6000 GPU, 24 GB memory). Runtimes shown here were obtained for one run for the rat dataset comprising 8 hours and 38 minutes of inertial

measurements (9.3M samples with a sampling rate of 30 Hz). The feature extraction, dimensionality reduction, clustering, and UMAP embedding steps are performed for the 44k segments resulting from the change-point detection step. Results are provided both as the total execution time on the dataset, excluding data loading time, and per hour of recording. (PDF)

**S5 Table. Computation times for the different steps of Keypoint-MoSeq**, measured on the same workstation used for running DeepLabCut, Anipose, and DISSeCT (Dell Precision 5820 Tower with an NVIDIA Quadro RTX 6000 GPU, 24 GB memory), and excluding data loading/formatting and PCA. Runtimes shown here were obtained for one run for the rat dataset comprising 8 h and 15 min of video recordings (891 052 frames recorded at 30 Hz). The number of iterations correspond to ones recommended by KpMS's online documentation (keypoint-moseq.readthedocs.io). Results are provided both as the total execution time on the dataset, excluding data loading time, and per hour of recording. (PDF)

**S6 Table. Post hoc examination of individual clusters (rat dataset).** For each cluster, the video snippets corresponding to the top 100 segments with the highest posterior probabilities of belonging to this cluster were examined, given a unique free label and linked with a unique main behavioral category among the following options: orienting, rearing, grooming, locomoting, idle or other (see Fig 1C and Methods). The table shows the first and second most frequent labels per cluster, sorting clusters according to their associated dominant category. For each cluster, the right column shows the percentage of segments belonging to this associated dominant category. The second and third columns from the left indicate the fraction of segments assigned to each cluster (% Segment) and the fraction of the total recording time that they represent (% Duration). (PDF)

**S1 Data. Data underlying Fig 3E.** This .csv file contains per-segment average head pitch and roll angles (see Methods). Each row corresponds to one segment. The first column (window_id) provides the unique segment identifier. The second and third columns (mu_pitch and mu_roll) contain the per-segment average head pitch and roll angles, respectively. The fourth column (cluster) indicates the cluster assignment of each segment. (CSV)

**S2 Data. Data underlying Fig 7C to 7E.** This .csv file contains the score under sham model (SUSM; see Methods) and the number of ipsiversive turns per minute for all mice across all experimental conditions. Each row corresponds to a single animal in a specific time bin. The first column (animal) provides the animal identifier. The second column (time_bin) specifies the period during which values were computed: "no L-DOPA" corresponds to weeks 2–4 (see Fig 7A); $t$−10 and $t$0 correspond to the last 10 minutes before and the first 10 minutes after L-DOPA injection, respectively; $t + 10$ corresponds to the interval from 10 to 20 minutes after injection, and so forth up to $t + 50$. The third column (group) indicates the experimental group as shown in Fig 7C and 7E. Note that mice from the PD group are reassigned to the PD_LDOPA group during the period immediately following L-DOPA injection (weeks 5–6, $t$0 to $t$50). The fourth column (SUSM) reports the SUSM values, and the fifth column (ipsi_turn) reports the number of ipsiversive turns per minute calculated over the specified period. (CSV)

**S1 Fig. Method for obtaining a gravity-aligned reference frame for 3D keypoint tracking data.** (A) Snout trajectory (black) obtained using Anipose, shown with the orientations of its principal components: PC1 (red), PC2 (green), and PC3 (blue). (B) The same trajectory after

transformation into a gravity-aligned reference frame. (C) Example 3D pose estimated using Anipose. (D) The same pose after transformation into the gravity-aligned reference frame. (TIFF)

**S2 Fig. Justification and tuning of the change-point detection (CPD) method.** (A) Video-based, frame-level human annotations (colored areas) and detected change points (vertical gray lines) overlaid on IMU data from the 14-second scratching episode shown in S1 Movie. Lowercase letters indicate the following behaviors: (a) sniffing with head pitched downward, (b) left body scratching, (c) left hind paw licking, (d) head scratching on the left side, and (e) decelerating head scratching on the left side. Change points were detected using Gaussian kernel-based Pruned Exact Linear Time (PELT) CPD applied to z-scored angular velocity ($\omega$) and gravitational acceleration ($\mathbf{a_G}$) data. Total acceleration ($\mathbf{a}$) is shown alongside $\mathbf{a_G}$ in lighter shades. (B) Number of detected change points (green) and median segment duration (purple) as a function of the penalty parameter ($\lambda$), computed from z-scored $\mathbf{a_G}$ and $\omega$ data in a 30-minute example recording. (C) Adjusted Rand Index (ARI) quantifying the similarity between segmentations obtained by running CPD on the full 30-minute recording versus running CPD in parallel on non-overlapping fixed-length windows. Notably, parallel CPD, which is computationally faster, produces nearly identical segmentations to the full-recording approach when window sizes are at least 5 minutes. (D) Histogram of mean angular speed values (i.e., the norm of gyroscope measurements) computed over non-overlapping, fixed-length windows of 700 ms. The x-axis is logarithmic to highlight the bimodal distribution, which reflects periods of immobility and movement. A threshold of 12 °/s on the standard deviation of angular speed was applied to isolate immobility periods. (E) Average log-likelihood (LL) computed across CPD segments or fixed-length windows from 10 independent GMM fits, plotted as a function of the number of mixture components $K$. LL values are shown with and without the ablation of immobility segments or windows prior to modeling (see main text for more details). For each $K$, the shaded region spans the full range of LL values across the 10 fits. Solid lines connect the maximum LL values at each $K$, indicating the best-fitting model. (TIFF)

**S3 Fig. Comparison of linear and Gaussian kernel change-point detection (CPD) methods.** (A) Average segment duration as a function of the penalty parameter ($\lambda$) for a 5-minute recording. Results are shown for linear CPD applied to the z-scored time-frequency representation of gyroscope data (green) and for Gaussian kernel CPD directly applied to z-scored gyroscope data (purple). As CPD is deterministic, it was run once for each type of kernel. Shaded areas represent 95% confidence intervals. (B) Adjusted Rand Index comparing the segmentations obtained from the two methods in panel A. (C) Example recording containing a wet dog shake (WDS) event. Top: Time-frequency representation of gyroscope signals, with the WDS' spectral signature highlighted by a dashed rectangle. Bottom: Corresponding gyroscope signals, with segmentation results from linear CPD on their z-scored time-frequency representation shown as green vertical lines. The time-frequency representation was computed using a Short-Time Fourier Transform (STFT) with a DPSS window (Length = 1 s, Half-bandwidth = 125 ms). Frequencies above 30 Hz were excluded. (D) Same recording as in C, with segmentation results from Gaussian kernel CPD on the z-scored gyroscope data shown as purple vertical lines. (TIFF)

**S4 Fig. Hyperparameter selection for GMM clustering of segment-wise features.** (A) Selection of the number of principal components for dimensionality reduction, based on

cross-validation error computed using the procedure described in [107]. The top panel shows cross-validation error as a function of the number of principal components; the bottom panel shows the corresponding fraction of explained variance. (B) Bayesian Information Criterion (BIC) values for increasing numbers of mixture components $K$ and different covariance matrix types (10 independent GMM fits per $K$ value and covariance matrix type). For each $K$, the shaded region spans the full range of BIC values across the 10 fits. Solid lines connect the maximum BIC values at each $K$, indicating the best-fitting model.
(TIFF)

**S5 Fig. Inertial and spectral signatures of an example wet dog shake event**. Traces of head gravitational acceleration ($\mathbf{a_G}$), non-gravitational acceleration ($\mathbf{a_{nG}}$), and angular velocity ($\boldsymbol{\omega}$) are shown. The bottom panel displays the log-magnitude of continuous wavelet transform coefficients computed on the yaw component of angular velocity. Vertical lines mark the start and end of the segment identified using Gaussian kernel-based PELT change-point detection (see S3 Fig).
(TIFF)

**S6 Fig. Comparison of 2D embedding methods for visualizing cluster distributions and segment-wise feature values.** (A) Columns (left to right): (1) Supervised UMAP (S-UMAP), incorporating both feature values and cluster identities; (2) Unsupervised UMAP using only feature values; (3) S-UMAP with shuffled cluster identities; and (4) t-SNE projection based on feature values (early_exaggeration = 12, perplexity = 30). Top row: segments colored by assigned cluster identity. Subsequent rows: segments colored by selected example feature values. (B) Cluster spread in the PC space (x-axis; trace of each cluster's covariance matrix) compared with spread in UMAP and S-UMAP embeddings (y-axis; average Euclidean distance of cluster members from their centroid), including either all clusters ($B_1$) or only non-sparse clusters ($B_2$), as defined in Fig 1C. (C) Pairwise inter-cluster distances in the PC space (x-axis; distances between centroids $\mu_i$ and $\mu_j$ for clusters $i$ and $j$, with $i>j$) compared with those in UMAP and S-UMAP embeddings (y-axis), including either all clusters ($C_1$) or only non-sparse clusters ($C_2$), as defined in Fig 1C.
(TIFF)

**S7 Fig. Identification of sparse clusters.** (A) Top: trace of the covariance matrix for each cluster, indicating cluster dispersion. The three highlighted clusters (in orange, green, and purple) correspond to the most dispersed clusters. Middle: percentage of the total number of segments contained in each cluster. Bottom: percentage of the total recording duration associated to each cluster. Clusters are sorted by ascending trace values in all panels. (B) Supervised UMAP embeddings displaying segments belonging to the three most dispersed clusters (in color) overlaid onto other segments (in gray).
(TIFF)

**S8 Fig. Grouping of similar leftward and rightward head movements by the categorical HMM.** Plots represent the joint visualization of the trajectories of $\mathbf{a_G}$ in the head reference frame (purple) and of the animal's heading in the Earth reference frame (orange) for the top 10 segments of pairs of clusters associated with four hidden states (see subsection 2.2 in S1 Text; RF: reference frame). Each pair represents the same head movement executed toward the left or right.
(TIFF)

**S9 Fig. Cluster transition matrix.** This graph represents the transition probability between clusters. Cluster numbers are ordered and color-coded according to their dominant main

behavioral category (Fig 1C). Colored squares highlight intra-category transitions.
(TIFF)

**S10 Fig. Main inter-category transitions.** (A) Left: chord plot highlighting state transitions corresponding to alternating rearing episodes and locomotor bouts. Right: Sankey diagram showing the proportion of state transitions occurring within isolated rearing episodes (blue) or within a full locomotion-rearing-locomotion sequence (green). The diagram is read from left to right. Colored vertical bars represent states, while shaded areas connecting states represent the proportion of state exits (right side of bars) and entries (left side of bars). (B) Left: chord plot highlighting state transitions corresponding to sequences during which the animal adopts a pitched-up head posture while keeping its front paws on the ground or close to it (low rear). Right: Sankey diagram showing the proportion of state transitions occurring in the context of this type of sequence (green).
(TIFF)

**S11 Fig. A categorical HMM enables the disambiguation of mixed clusters.** (A) Posterior probability matrix, displaying the likelihood of each state given each cluster, denoted as $P(state|cluster)$. Arrowheads highlight two example clusters (12 and 19) that are associated with multiple states. (B) Top: Bar plot illustrating the posterior probabilities for cluster 12, showing a strong association with states 2 and 8. Middle: Pie charts displaying the proportion of unique labels assigned to the top 100 segments with the highest $P(state = 2|cluster = 12)$ (left) or $P(state = 8|cluster = 12)$ (right). Note that while the same labels appear in both groups, their proportions differ drastically. Bottom: joint visualization of the trajectories of $\mathbf{a_G}$ in the head reference frame (purple) and of the animal's heading in the Earth reference frame (orange) for the top 30 segments in each group. (C) Top: Bar plot showing the posterior probabilities for cluster 19, indicating a predominant association with states 3 and 11. Middle: Pie charts displaying the proportion of unique labels given to the top 100 segments with the highest $P(state = 3|cluster = 19)$ (left) or $P(state = 11|cluster = 19)$ (right). Note that labels differ between the two groups, although the same pattern fills are used. Bottom: Joint visualization of the trajectories of $\mathbf{a_G}$ in the head reference frame (purple) and of the animal's heading in the Earth reference frame (orange) for the top 30 segments in each group.
(TIFF)

**S12 Fig. Comparison of the output of DISSeCT and Keypoint-MoSeq (KpMS).** (A) Distribution of segment durations for DISSeCT and syllable instance durations for KpMS, shown as frequency histograms (left) and empirical cumulative distribution functions (ECDFs; right). Median durations are indicated above the histograms. (B) Left: cumulative time coverage—i.e., the cumulative proportion of total recording time—as a function of the number of clusters (DISSeCT) or syllables (KpMS), sorted by decreasing contribution. A minimum of 23 KpMS syllables (vertical orange dashed line) and 36 DISSeCT clusters (vertical blue dashed line) were required to account for at least 95% of the total recording time (horizontal black dashed line). Right: cumulative frequency—i.e., cumulative proportion of total instances or segments—as a function of the number of syllables or clusters, sorted by decreasing contribution. At least 23 KpMS syllables and 43 DISSeCT clusters were needed to account for 95% of all instances and segments, respectively. (C) Stacked bar plots showing the fraction of segments in each DISSeCT cluster assigned to different behavioral categories, based on manual inspection of either the top 100 segments per cluster (top; same data as in Fig 1C) or a random sample (bottom; see Methods). (D) Same as C, but for KpMS syllables. Top: top 100 instances per syllable. Bottom: randomly selected instances (see Methods). (E) Purity scores for each DISSeCT cluster (left; excluding sparse clusters) and KpMS syllable (right), based

on either random samples (*x*-axis) or the top 100 segments/instances (*y*-axis). Each dot represents a cluster or syllable, colored by its dominant behavioral category (as in C-D). The solid diagonal line indicates identity; dashed lines denote $\pm 10\%$ deviation. (F) Fraction of clusters or syllables with a purity score above a given threshold, plotted as a function of that threshold. Solid lines: top 100 segments/instances. Dashed lines: random samples. (G) DISSeCT vs. KpMS purity scores at the behavioral category level. Each behavioral category was associated with a specific set of clusters/syllables, based on the dominant label among their top 100 segments/instances (top panels in C and D). Purity scores were then computed by aggregating the labels of these segments/instances, using either the top 100 most representative ones per cluster/syllable (light colors) or a random global selection (dark colors; see Methods). **H**, Stacked bar plots showing label assignments for the top 100 instances of each grooming-related KpMS syllable.
(TIFF)

**S13 Fig. Comparison of cluster and syllable assignment in DISSeCT vs. KpMS.** For this analysis, KpMS syllables and DISSeCT clusters accounting for less than 5% of the total recording duration were excluded. DISSeCT outlier segments and sparse clusters were also excluded. (A) Row-normalized contingency matrix showing the conditional probability of DISSeCT clusters given KpMS syllables. Both syllables and clusters are grouped by behavioral category. (B) Same analysis at the behavioral category level. (C) Column-normalized contingency matrix showing the conditional probability of KpMS syllables given DISSeCT clusters. (D) Same analysis at the behavioral category level.
(TIFF)

**S14 Fig. Distribution of segments in 2D embeddings of the standardized feature space for sham and lesioned mice.** This figure shows the results of two fully unsupervised embedding methods (UMAP: top; t-SNE: bottom) and one semi-supervised method (S-UMAP: middle), applied to Sham (left, blue) and PD mice before (middle, green) and during (right, orange) L-DOPA treatment. For S-UMAP, cluster identities were derived from a Gaussian Mixture Model (GMM) fitted on data from sham mice. Hyperparameters were set as follows: n_neighbors = 8, n_components = 2 and min_dist = $1 \times 10^{-3}$ for both UMAP and S-UMAP, as well as target_weight = $1 \times 10^{-9}$ for S-UMAP; early_exaggeration = 12 and perplexity = 30 for t-SNE.
(TIFF)

**S1 Movie. Example recording illustrating IMU data segmentation using change-point detection.** Left: Gravitational ($\boldsymbol{a_G}$) and non-gravitational ($\boldsymbol{a_{nG}}$) components of head acceleration, alongside head angular velocity ($\boldsymbol{\omega}$), for an 8-second self-grooming episode. Alternating color bands mark segment boundaries. Notably, rapid changes in head orientation appear as peaks in the angular velocity trace and are accurately identified as distinct segments by the change-point detection algorithm. Right: Simultaneous video recording from beneath a glass floor. IMU and video recordings are synchronized and displayed at half speed.
(MP4)

**S2 Movie. 2D visualization of per-segment head attitude and heading trajectories.** This video details the method used to generate a joint visualization of the trajectories of $\boldsymbol{a_G}$ in the head reference frame (purple) and of the animal's heading in the Earth reference frame (orange) for a series of segments (here belonging to the same cluster). See Methods.
(MP4)

**S3 Movie. Example recording illustrating IMU signals during mobile olfactory exploration.** Left: Non-gravitational component of head acceleration ($\boldsymbol{a_{nG}}$) and head angular

velocity ($\omega$) for a 2-second episode of mobile olfactory exploration (see also Fig 4). Right: Simultaneous video recording. IMU and video recordings are synchronized and displayed at half speed.
(MP4)

**S4 Movie. Animated gif showing the representative keypoint trajectories for each syllable projected onto the horizontal (*x,y*) plane, generated by the `get_typical_trajectories` function of Keypoint-MoSeq.**
(GIF)

**S5 Movie. Animated gif showing the representative keypoint trajectories for each syllable projected onto the sagittal (*x,y*) plane, generated by the `get_typical_trajectories` function of Keypoint-MoSeq.**
(GIF)

## Author contributions

**Conceptualization:** Clément Léna, Daniela Popa, Pierre Latouche, Guillaume P Dugué.

**Data curation:** Romain Fayat.

**Formal analysis:** Romain Fayat.

**Funding acquisition:** Clément Léna, Daniela Popa, Guillaume P Dugué.

**Investigation:** Romain Fayat, Marie Sarraudy.

**Methodology:** Romain Fayat, Pierre Latouche, Guillaume P Dugué.

**Project administration:** Pierre Latouche, Guillaume P Dugué.

**Resources:** Romain Fayat.

**Software:** Romain Fayat, Pierre Latouche.

**Supervision:** Pierre Latouche, Guillaume P Dugué.

**Validation:** Romain Fayat, Guillaume P Dugué.

**Visualization:** Romain Fayat, Guillaume P Dugué.

**Writing – original draft:** Romain Fayat.

**Writing – review & editing:** Clément Léna, Daniela Popa, Pierre Latouche, Guillaume P Dugué.

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
