## [Editor Report · Decision Letter 0]

16 Nov 2024

Dear Dr Dugué,

Thank you for submitting your manuscript entitled "Fine decomposition of rodent behavior via unsupervised segmentation and clustering of inertial signals" for consideration as a Methods and Resources by PLOS Biology.

Your manuscript has now been evaluated by the PLOS Biology editorial staff, as well as by an academic editor with relevant expertise, and I am writing to let you know that we would like to send your submission out for external peer review.

Once your full submission is complete, your paper will undergo a series of checks in preparation for peer review. After your manuscript has passed the checks it will be sent out for review. To provide the metadata for your submission, please Login to Editorial Manager (https://www.editorialmanager.com/pbiology) within two working days, i.e. by Nov 20 2024 11:59PM.

Kind regards,

Taylor

Taylor Hart, PhD,

Associate Editor

PLOS Biology

thart@plos.org

---

## [Decision Letter · Decision Letter 1]

27 Jan 2025

Dear Dr Dugué,

Thank you for your patience while your manuscript "Fine decomposition of rodent behavior via unsupervised segmentation and clustering of inertial signals" was peer-reviewed at PLOS Biology. It has now been evaluated by the PLOS Biology editors, an Academic Editor with relevant expertise, and by several independent reviewers.

In light of the reviews, which you will find at the end of this email, we would like to invite you to revise the work to thoroughly address the reviewers' reports.

As you will see below, the reviewers think that paper reports a method that appears useful and has wide potential interest. However, R2 and R3 raise concerns about how DISSeCT is validated, especially the lack of direct comparison with video-based tracking methods. The limited level of validation reduces the argument for widespread adoption of the new method. All reviewers, but especially R2 and R3 asked questions about various technical aspects of the method and suggested new analyses.

A revised manuscript will need to include a quantitative comparison between DISSeCT and video-based tracking approaches, as well as an expanded discussion of the method's generalizability, use cases and limitations. You will also need to address the technical concerns and questions raised by the reviewers.

Given the extent of revision needed, we cannot make a decision about publication until we have seen the revised manuscript and your response to the reviewers' comments. Your revised manuscript is likely to be sent for further evaluation by all or a subset of the reviewers.

**IMPORTANT - SUBMITTING YOUR REVISION**

*Re-submission Checklist*

*Published Peer Review*

*PLOS Data Policy*

*Blot and Gel Data Policy*

Sincerely,

Taylor

Taylor Hart, PhD,

Associate Editor

PLOS Biology

thart@plos.org

REVIEWS:

Reviewer #1:

The paper effectively describes the use of an unsupervised learning pipeline to analyse rodent behavior from IMU data. The authors present compelling arguments for the usefulness of this approach as compared to multi-angle, video-based pose estimation techniques. Their pipeline appears both sensible and robust, and the accompanying data and GitHub repository are well-formatted and easy to follow, making this method readily adoptable by other laboratories.

From a computational standpoint, while none of the unsupervised techniques employed are individually novel, the overall demonstration of their applicability to IMU data is indeed novel, valuable, and likely poised for widespread adoption.

We do raise one concern regarding the generalizability of their model, along with a few suggestions for additional statistical tests that would further strengthen the applicability of their technique. However, we want to emphasise that the work is already suitable for acceptance as is. These suggestions are meant as optional considerations that the authors may wish to discuss or incorporate at their/editorial discretion.

One discussion point is about how generalizable the learnt parameter space is. It would be convincing to run the original model on a new ethological dataset (e.g. mice recorded in a distinct complex enriched environment) to perform a comparison of the original model and the new model trained solely on the novel dataset. They could compare the model likelihoods using an F-test, Wilks test, or a Bayesian information criterion measure.

A second comment is about demonstrating more convincingly the usefulness of their change-point detection method. It would be sufficient to show that this creates statistically better features for their clustering method than a random segmentation of the dataset into arbitrary time windows, so the authors would just need to retrain their PCA/GMM model on these random segments, and perform statistical tests to show the likelihood is statistically significantly worse on the random features.

Finally, to be more convinced of the Parkinson's study, they could once again train a model on the mice after L_DOPA administration, and compare this model likelihood with the sham model, to show a significant difference in likelihoods from the two models on the Parkinson's model animals, thus demonstrating statistically a difference in behaviours between the sham and diseased animals.

As a more general comment, I don't find it that clear in the discussion whether the authors are encouraging the use of their pipeline, or their actual learnt model parameters. The latter case will take more convincing of its broader applicability, while the former makes a lot of sense as a clustering pipeline, but would lose the detailed annotations of the clusters they found for their particular dataset, as clusters from a new model can't necessarily be identified with the clusters found in their model. More clarity on this point would be beneficial.

A few more minor comments:

Could they validate the use of their kernel vs other kernels in their change-point detection cost function?

Why does this loss function include the length of each segment, when, I believe, the total loss will therefore just have the length of the recording, which is a constant, so this term is unnecessary?

Why did they use the Viterbi filter over the Baum-Welch algorithm (which should give more robust results) to train their HMM?

Reviewer #2: The study offers an entirely IMU-based approach for fine-grained, unsupervised parsing of rodent behavior (DISSeCT) that bypasses the usual computer-vision pipeline. Overall, the paper seems to be useful for behavior segmentation.

### Specific Comments

0. Please add background information on current unsupervised methods. How is this method different (in addition to just being applicable to IMU data).

1. Comparisons with other unsupervised methods

 1a. In Keypoint-MoSeq, segmentation and clustering are integrated in a hierarchical Bayesian model that simultaneously infers where a behavioral "syllable" starts and ends, as well as which syllable it is. In the present paper, segmentation is done first via kernel-based change-point detection (Section "Pipeline architecture"), and then the extracted segments are clustered using a GMM. There is no direct demonstration of whether a fully joint approach would outperform or produce different motifs.

 1b. VAME uses a temporal VAE on pose data to learn embeddings of short windows of motion, then clusters those embeddings (sometimes with an HMM). Again, because DISSeCT does not embed data via a neural network, it avoids the potential complexity and resource overhead of a deep VAE, but it might lose out on the automatic feature learning and nonlinear features that VAME provides. Instead, the authors rely on manually crafted features (240 of them) and then do PCA + GMM. A big question is whether a more data-driven and nonlinear embedding (VAE) might capture often-subtle IMU patterns that the curated features do not.

2. The current study validates cluster assignments by having an observer label ~ N video snippets per cluster, and then compute a "consistency score" if a single label dominates (Figure 1F). While that shows approximate coherence, it is effectively a manual annotation approach. Other standard metrics in unsupervised learning include purity, normalized mutual information, or silhouette scores. Surprisingly, the authors do not report any such measures. From a machine learning perspective, the argument that "consistency score" suffices feels incomplete—particularly because they are claiming fully unsupervised segmentation that "largely aligns" with human-labeled behaviors. Reporting something like purity or ARI (adjusted Rand index) would make the approach more quantitatively comparable both internally (e.g. across hyperparameters) and to other frameworks (Keypoint MoSeq, VAME).

3. The GMM is used to derive 48 clusters (p5-6, Fig. S4B) I see they use the BIC across different numbers of clusters, but the text conflates "48 mixture components" with "48 clusters," and it's important to confirm whether each mixture component actually maps one-to-one to a single behavioral cluster. Especially since they note that some clusters are "sparse" (Fig. S7) or contain ambiguous segments. If multiple mixture components are effectively merged later (e.g. through the HMM or operator inspection), the text should explicitly state how the final cluster identities are consolidated.

4. I'm also confused about the logic for using a categorical HMM only at the end (Section "Behavioral patterns can be unveiled...", p.10). The results show that this HMM can disambiguate context-specific segments that cluster similarly in GMM space. But why not incorporate the temporal constraints (i.e., HMM) into the initial segmentation process, akin to Keypoint-MoSeq or MoSeq? In other words, is the pipeline simpler or faster by strictly separating change-point detection, GMM clustering, and then HMM post-processing? The authors claim this keeps things modular, but I would be curious if an ablation test shows how final motif quality might differ if you skip the HMM or if you incorporate it jointly before segmentation.

5. Many animal behaviors can be nested at multiple timescales (e.g., whisking vs. grooming vs. longer bouts of rearing). It is unclear whether a single, universal penalty λ captures them all, or if the user might need a multi-level approach for more complex data. This concern is not specific to DISSeCT: Keypoint MoSeq's HMM-based segmentation also implicitly fixes a timescale, and VAME's window-based embeddings have an assumed window size. None of these methods, as far as I know, truly adapt to multiple timescales in a single pass. On the practical side, additional guidelines (e.g., instructions on how to choose λ based on, say, user-defined expectations or frequency-domain analyses) would help experimentalists who do not want to rely solely on a published timescale from a different context.

6. The entire pipeline is validated on single-animal data in controlled arenas, which is valuable from a proof-of-concept standpoint but sidesteps complexities of multi-animal interactions. DISSeCT's reliance on a single head-mounted IMU presumably makes it tricky to separate signals if multiple animals interact or physically collide. In Keypoint MoSeq or VAME (which are typically video-based pose estimators), you at least attempt multi-animal identity tracking, though that is itself non-trivial. The authors should at least have a discussion of this issue towards the end of the paper.

7. Keypoint-based pipelines sometimes show reduced performance on out-of-sample populations whose size, coat color, or posture deviates from the training distribution. In principle, an IMU-based approach may be somewhat more robust to superficial morphological differences (as inertial signals can still pick up speed, angle changes, etc.). However, if subtle morphological or gait differences shift the distribution of features significantly, a newly collected dataset might need re-segmentation parameters (especially the penalty λ) or a newly fitted GMM. As it stands, the paper does not mention cross-genotype validations or how DISSeCT might be retrained or adapted for genetically diverse cohorts. It would be helpful if the authors clarified whether the current pipeline can "auto-tune" itself (using BIC or cross-validation for number of clusters) when the population differs profoundly in mass, limb length, or baseline movement speed.

Overall, I think the pipeline is strong in capturing time-varying inertial signals and addresses a clear gap by providing a flexible, purely IMU-based segmentation approach. However, I would have liked a direct quantitative comparison with Keypoint MoSeq, VAME on a shared dataset, and with at least a standard cluster metric (e.g. purity, ARI, mutual information). Right now, the assessment hinges heavily on manual annotation of top-segment video snippets. That approach works, but it makes it harder to compare to existing frameworks or to interpret the "48 clusters" as truly distinct. From a machine learning perspective, I'd suggest adding standard unsupervised metrics, clarifying parameter sensitivity, and possibly discussing how a deeper (VAE-based) embedding might integrate with their kernel-based segmentation method. Otherwise, the method remains an interesting alternative to camera-based pose estimation, especially in situations where cameras aren't feasible or where inertial readings offer finer kinematic signals at minimal computational cost.

Reviewer #3: The authors present a technique for quantitative behavioral phenotyping in rats that uses a head-mounted IMU to bypass the need for cameras and video-based tracking. The method is particularly appealing because it's possible experimenters already mounting drives or miniscopes on the head could simply add an IMU and get relatively comprehensive behavioral phenotyping "for free" (although the authors did not explicitly test their system as a component of an ephys or imaging system). The authors show that they can segment and cluster dozens of different rat behaviors given only a small set of head inertial measurements. They end with an experiment showing that their measurements and analyses are sufficient to detect levodopa-induced dyskinesias in a 6-OHDA model of PD.

I really liked this paper, and it was a joy to read. The authors communicate clearly and have made beautiful, informative figures. I think the method is exciting and impactful, and I look forward to seeing it published. However, I did have several questions, comments, and concerns that I think, if addressed, would help readers gain a better sense of the approach and its limits (and thus gain more trust in the method and its potential power).

Note, there are situations where using IMUs/DISSeCT would likely be far superior to video-based techniques. Not just in efficiency but in the ability to make measurements at all. Namely, when there are tons of occlusions in the environment (social groups, especially large groups; more natural arenas with lots of objects or housing; outside in the field, etc.) Obviously one can't do everything in a single paper, but if the authors wanted to elevate the work in the future, doing these types of experiments would really showcase the unique power of the technique.

Questions, comments, concerns:

1. I am confused about how the authors incorporated 3D pose data into their analyses. As best as I can tell, the poses are used only for visualizing example traces of their IMU clusters (and tracking the tail base velocity, which doesn't require tracking a full 3D pose). If just for visualization and qualitative validation, why do you need to track pose at all? This could have just been done with raw, untracked video.

2. I was expecting to see the 3D pose used in a different, quantitative way to validate their IMU clusters and defend their statements about the advantages of IMUs over (and as potential replacement for) video tracking approaches. Behavioral clustering from video is the current state-of-the-art, so it is important to compare directly to it.

3. Related to (2), you seem to find fewer clusters than previous studies quantifying rat behavior with 3D pose tracking. And potentially the clusters you find are less consistent (although cluster consistency is not often explicitly computed in other studies - I applaud the authors for doing this, though I do raise questions, below, about how they did it).

4. Aren't your consistency values going to be inflated and not representative of a cluster's quality overall if you are labeling only the top 200 posterior samples? I assume there are many more than 200 samples in a cluster, and that the samples with the lowest posterior probability are going to be the least consistent.

5. More details on the consistency score, please. Specifically how are the no video / less than 7 frames segments accounted for - shouldn't they be integrated into your consistency? For instance, cluster 29 seems mostly consistent, but then you see that only a small fraction were valid behavior segments, so I would call that overall inconsistent, no?

6. Can you provide some guidance on how to interpret the consistency values? To me they seem kind of low, but perhaps I don't have a good baseline. What do consistency scores look like for pose tracking clusters? Why not try this comparison if you have the data?

7. What happens if you cluster based on pose and then looking at distributions of head kinematics? Or overlay pose cluster ID onto your IMU S-UMAP?

8. Did you try an ARHMM directly on the head kinematic measurements? Would be too much to ask to implement, but mention of its possible advantages/disadvantages would be great.

9. I think the supervised UMAP (S-UMAP) visualizations are a little misleading, as they give the false impression of there being super distinct clusters based on features alone (i.e., with an unsupervised UMAP). It is particularly misleading because most everyone in this space is using unsupervised embeddings, so it produces this false sense that the IMU data is far superior. What about also including unsupervised UMAP in the main figure? This actually would be a nice tie-in to previous work, which would allow readers to better understand your study in the context of previous work.

10. Are there analyses you could do to better understand how the cluster IDs are influencing the S-UMAP space? What does it look like if you run GMM on a random matrix and then embed with S-UMAP? Or for a shuffled version of your data?

11. If the S-UMAP structure is primarily driven by cluster ID, what purpose do the S-UMAP visualizations serve that couldn't instead be communicated with a heatmap of the features, or equivalent?

12. For mixed grooming clusters and the like, kinematic ambiguities were mentioned as a possible cause. Can the authors reflect on whether they have evidence that the issues stem from the measurements and not the clustering approach? Because if it's the clustering approach, that means the problem could potentially be solved with future iterations or advances in the clustering, or with application of alternative existing clustering approaches (in a future study).

13. In your PD experiment analyses, sham is always going to be better fit across all time points, by virtue of the sham animals having been the ones used to fit the GMM. Have you tried fitting the GMMs to a subset of the sham? You could do this with a cross-validation approach (say, leave-one-out) and analyze each sham animal when it is in the "test" set, to better match how the PD condition is analyzed.

14. For the LID result, again it would be nice to know more about the interplay between the supervised and unsupervised components. Is the erroneous cluster identity over-accentuating the actual kinematic differences between the LID and sham data? There could be relatively small differences in the underlying features that lead to a change in hard cluster assignment to the sham-GMM - would this then produce misleadingly large differences in the S-UMAP embedding? What does it look like if you do unsupervised clustering alone?

15. Can the authors comment on why the PD phenotype (pre levodopa) was not obvious from the S-UMAP maps?

16. I am curious about the 3 LID animals that look more like sham and whether there could be any sort of batch/cohort effect at play here. Were the sham animals run in a separate batch (e.g., different litter or a long time apart or a different experimenter or a different IMU installation point, etc)? Is there a chance, say, that something unrelated to LID is driving the observed difference? Example: the 9 animals with the clear LID effect are from a different litter, or were performed at different time of day, and just happened to become more lethargic or habituated more to the arena over time.

17. I would suggest an expanded discussion of possible limitations in the text. For example, the invasiveness of the IMU vs. video tracking; scalability of the procedure (requires a surgical procedure, whereas video tracking doesn't); reduced interpretability compared to pose tracking (you showed poses to aid with interpretability in the paper, after all). Just curious, have you ever tried to regress poses from inertial variables?

Relatively minor:

1. Will their IMU fit on a mouse? Potential discussion point.

2. L69: "artifact-free" is a strong claim and can't possibly be true - can you just stick with the claim of "low noise"?

3. L112: "different shapes" - gives the impression these shapes can be arbitrary, but they still have to be Gaussian (just not spherical Gaussian).

4. L116: BIC.. can also be used with K-means. K-means is, after all, just a special case of a GMM. Also, did you try a cross-likelihood analysis? BIC is notoriously unreliable.

5. Missing citation and discussion of Ouyang et al. 2024 Neuron, which uses IMUs to segment behaviors (although not as extensively as your paper does).

6. I found myself wanting to know earlier on in the paper (say, in the Introduction), exactly how many / what type of IMUs were being used.

7. No citation of Fig. 2D_1 or Fig. 2D_2 in the text.

8. L186 paragraph - Found this paragraph confusing, it's written as if you've already described the segmentation of different locomotion phases (which I interpreted to mean different components of the gait cycle). If by "phases" you mean periods when the animal is locomoting, then I suggest a different term to avoid ambiguity. Also, I personally find the statement about (paraphrasing) "because there is no location information one shouldn't necessarily expect to identify locomotion segments" a weak straw man - I don't think anyone would expect the head to be perfectly still during locomotion.

9. L358 - I don't think you know whether you've captured the "the full repertoire" of head movements - there could be many behaviors you're not observing, especially when just using an unnatural open field box. Do you need to claim "the full"?

10. L397 - Re: clustering after dimensionality reduction to 2d, "its drastic dimension reduction step prior to clustering likely diminishes its discriminative power [70]". "Likely" is an overstatement. Something like "potentially" would be better. The study you cite is on drosophila behavioral clustering, and you've made no direct comparisons to clustering in 2D (although you could…).

11. L427 - "DISSeCT also meets the requirements of a robust deep phenotyping tool." I think it is fine to say it holds promise, but I wouldn't say you've "met the requirements" yet. First, what are the requirements? How did you explicitly test for robustness? It didn't detect differences in all animals, so that seems to be a knock against "robustness." Also, the detected changes in PD (pre-L-DOPA) were relatively small, despite the 6-OHDA model being quite a strong perturbation, you did not test it in subtler PD models, did not detect a positive L-DOPA rescue of a PD phenotype, etc. I'm not saying you needed to have done all of this, but the claim you make here should reflect how there is probably more needed to "meet the requirements".

---

## [Decision Letter · Decision Letter 2]

2 Sep 2025

Dear Dr Dugué,

Thank you for your patience while we considered your revised manuscript "Fine decomposition of rodent behavior via unsupervised segmentation and clustering of inertial signals" for publication as a Methods and Resources at PLOS Biology. This revised version of your manuscript has been evaluated by the PLOS Biology editors, the Academic Editor, and one of the original reviewers.

Based on the reviews, we are likely to accept this manuscript for publication, provided you satisfactorily address the remaining points raised by the reviewer (see the end of this email). Please also make sure to address the following data and other policy-related requests.

IMPORTANT: In this next revision, you should address both the remaining reviewer comments and the following editorial requirements:

------------------

**Title

We suggest the following alternative title for your study, to highlight your method's name and maintain accessibility for our broad audience:

"DISSeCT is an analytical framework for high-resolution rodent behavior mapping using inertial sensors"

**Competing interests:

The first character in the word 'the' is missing from your competing interest statement.

**Ethics:

The Ethics statement needs to be the first subheading in the Material & Methods section. It must include the full name of the IACUC/ethics committee that reviewed and approved the animal care and use, as well as the protocol/permit/project license number. https://journals.plos.org/plosbiology/s/ethical-publishing-practice

**Data:

Your data availability statement says that the data will be made accessible upon acceptance of the manuscript. As your paper is likely to be formally accepted, please make the data accessible now and provide a link.

In addition, please note that we require the numerical values in a supplementary file or as a permanent DOI'd deposition for the following figures:

3E

7C

Please cite the location of the data clearly in all relevant main and supplementary Figure legends, e.g. “The data underlying this Figure can be found in S1 Data” or “The data underlying this Figure can be found in https://doi.org/10.5281/zenodo.XXXXX”

If you are reporting experiments where n ≤ 5, please plot each individual data point.

Supplementary files (e.g., excel). Please ensure that all data files are uploaded as 'Supporting Information' and are invariably referred to (in the manuscript, figure legends, and the Description field when uploading your files) using the following format verbatim: S1 Data, S2 Data, etc. Multiple panels of a single or even several figures can be included as multiple sheets in one excel file that is saved using exactly the following convention: S1_Data.xlsx (using an underscore).

**Code availability:

Thank you for providing the underlying code in GitHub. However, because Github depositions can be readily changed or deleted, please make a permanent DOI’d copy (e.g. in Zenodo) and provide this URL in the manuscript and Data Availability Statement.

-----------------

We expect to receive your revised manuscript within two weeks.

*Published Peer Review History*

*Press*

Sincerely,

Taylor

Taylor Hart, PhD,

Associate Editor

thart@plos.org

PLOS Biology

Reviewer remarks:

Reviewer #3: I have reviewed the revised manuscript and the authors' responses to all reviewers. The authors were impressively responsive to reviewer feedback and have performed many new analyses that I think satisfactorily address all major reviewer concerns.

I have a few additional comments:

Re: Reviewer 2, comment 7. The authors added to the discussion section a recommendation that the pipeline should be rerun whenever the experimental model and conditions change. However, there are clear disadvantages to this that are also important to mention in the discussion (along with communicating why it might be important to address these issues in future work). Also, with pose data, if the data from two experiments is not drastically different, it is common to combined and embed the datasets together. I think it is worth mentioning under which conditions this might work for IMU data and your model, rather than simply saying researchers should always train a new model.

Re: Reviewer 3, comments 9-10. I remain unconvinced of S-UMAP utility. The authors state that S-UMAP enhances interpretability or "more clearly reveals the internal structure" or clusters or provides a more interpretable "intra-cluster feature consistency and inter-cluster boundaries", but just because something looks nicer doesn't mean it's more accurate and reflective of underlying trends. What is the quantitative evidence that clusters being larger or smaller in S-UMAP reflects the underlying variance of those clusters. That the distance between S-UMAP clusters is indeed indicative of cluster similarity? Much of this could be imposed or influenced in unmeaningful ways by the supervision process.

However, I do appreciate that the authors have in their revision now used unsupervised UMAP where it is more essential to do so, to support their more essential claims related to PD.

Overall, great work. I hope you feel the reviews have been constructive.

---

## [Editor Report · Decision Letter 3]

22 Sep 2025

Dear Dr Dugué,

Thank you for the submission of your revised Methods and Resources "DISSeCT: an unsupervised framework for high-resolution mapping of rodent behavior using inertial sensors" for publication in PLOS Biology. On behalf of my colleagues and the Academic Editor, Eric Nestler, I am pleased to say that we can in principle accept your manuscript for publication, provided you address any remaining formatting and reporting issues. These will be detailed in an email you should receive within 2-3 business days from our colleagues in the journal operations team; no action is required from you until then. Please note that we will not be able to formally accept your manuscript and schedule it for publication until you have completed any requested changes.

PRESS

Sincerely, 

Taylor Hart, PhD,

Associate Editor

PLOS Biology

thart@plos.org